# Discovery of beta-lactamase CMY-10 inhibitors for combination therapy against multi-drug resistant Enterobacteriaceae

**Nousheen Parvaiz[1‡], Faisal Ahmad[1‡], Wenbo Yu[2‡], Alexander D. MacKerell, Jr.[2]\*, Syed Sikander Azam[1]\***

**1** Computational Biology Lab, National Center for Bioinformatics, Quaid-i-Azam University, Islamabad, Pakistan, **2** University of Maryland Computer-Aided Drug Design Center, Department of Pharmaceutical Sciences, School of Pharmacy, University of Maryland, Baltimore, MD, United States of America

‡ These authors contributed equally and can be considered as first author.
\* syedazam2008@gmail.com (SSA); alex@outerbanks.umaryland.edu (ADM)

**Data Availability Statement:** All relevant data are within the manuscript and its Supporting Information files.

## Abstract

β-lactam antibiotics are the most widely used antimicrobial agents since the discovery of benzylpenicillin in the 1920s. Unfortunately, these life-saving antibiotics are vulnerable to inactivation by continuously evolving β-lactamase enzymes that are primary resistance determinants in multi-drug resistant pathogens. The current study exploits the strategy of combination therapeutics and aims at identifying novel β-lactamase inhibitors that can inactivate the β-lactamase enzyme of the pathogen while allowing the β-lactam antibiotic to act against its penicillin-binding protein target. Inhibitor discovery applied the Site-Identification by Ligand Competitive Saturation (SILCS) technology to map the functional group requirements of the β-lactamase CMY-10 and generate pharmacophore models of active site. SILCS-MC, Ligand-grid Free Energy (LGFE) analysis and Machine-learning based random-forest (RF) scoring methods were then used to screen and filter a library of 700,000 compounds. From the computational screens 74 compounds were subjected to experimental validation in which β-lactamase activity assay, in vitro susceptibility testing, and Scanning Electron Microscope (SEM) analysis were conducted to explore their antibacterial potential. Eleven compounds were identified as enhancers while 7 compounds were recognized as inhibitors of CMY-10. Of these, compound 11 showed promising activity in β-lactamase activity assay, in vitro susceptibility testing against ATCC strains (*E. coli*, *E. cloacae*, *E. agglomerans*, *E. alvei*) and MDR clinical isolates (*E. cloacae*, *E. alvei* and *E. agglomerans*), with synergistic assay indicating its potential as a β-lactam enhancer and β-lactamase inhibitor. Structural similarity search against the active compound 11 yielded 28 more compounds. The majority of these compounds also exhibited β-lactamase inhibition potential and antibacterial activity. The non-β-lactam-based β-lactamase inhibitors identified in the current study have the potential to be used in combination therapy with lactam-based antibiotics against MDR clinical isolates that have been found resistant against last-line antibiotics.

**Funding:** This study was funded by the Pakistan-United States Science and Technology Cooperation Program SSA and ADM (US/2017/360) and the USA National Institutes of Health to ADM (GM131710). The funders had no role in study design, data collection and analysis, decision to publish, or preparation of the manuscript.

**Competing interests:** The authors have declared that no competing interests exist. ADM is co-founder and CSO of SilcsBio LLC. This does not alter our adherence to PLOS ONE policies on sharing data and materials.

## Introduction

Despite the incredible initiatives taken in modern medicine to combat antibiotic-resistant microorganisms, emerging β-lactamase-mediated resistance remains a threat to the most prominent class of antibiotics, the β-lactams [1]. According to report by World Health Organization (WHO), urgent action is required to combat antimicrobial resistance expected to cause a global financial crisis by forcing 24 million people into extreme poverty by 2030 and causing 10 million deaths annually by 2050 [2]. The most problematic multi-drug resistant microorganisms are Gram-negative pathogens by acquiring mobile genetic elements linked with multiple resistance factors for most antibacterial agents [3]. Literature evidence strongly suggests that Enterobacter species such as *Enterobacter cloacae*, *Enterobacter agglomerans*, *Enterobacter alvei*, and *Enterobacter aerogenes* characterized by potential antimicrobial resistance mechanisms are one of the leading causes of fatal nosocomial infections worldwide [4].

The production of β-lactamases by Gram-negative pathogens which can hydrolytically inactivate β-lactam drugs is a prevalent resistance mechanism that renders even the most effective antibiotics ineffective. According to the molecular classification, β-lactamases are divided into class A, B, C, and D enzymes. β-lactamase enzymes of class A, C, and D use serine to hydrolyze β-lactam ring. However, β-lactamases of class B are metalloenzymes which use divalent zinc ions to hydrolyze the substrate. Research states that class C β-lactamases have conferred resistance against β-lactam containing β-lactamase inhibitors in combination with the clinically used β-lactamase inhibitor clavulanate [5]. Plasmid encoded class C β-lactamases are more problematic compared to other classes because of horizontal gene transfer [5]. The emergence of carbapenemase and Extended-spectrum β-lactamase (ESBL) producing Enterobacteriaceae, which are resistant even to third-generation antibiotics, pose a serious therapeutic challenge worldwide. The current study, therefore, targets the β-lactamase CMY-10 to design novel and effective inhibitors that can reestablish antibiotic activity against bacteria producing serine β-lactamases.

β-lactam antibiotics act as substrate mimics of the penultimate d-Ala-d-Ala on the peptidoglycan stem peptide targeting the final step in peptidoglycan synthesis and inhibiting the transpeptidation of adjacent peptidoglycan strands [1]. Inhibitors such as tazobactam, clavulanic acid, and sulbactam containing β-lactam ring were previously used to overcome serine β-lactamase-mediated resistance [1]. These inhibitors themselves contain s lactam ring and are structurally similar to β-lactam antibiotics. They target β-lactamase by creating stable acyl-enzyme intermediate in the active site with the catalytic serine. One of the downsides of using structurally similar β-lactamase inhibitors in combination therapy is that in most cases initial competitive binding between antibiotics and inhibitors to β-lactamase results in significant loss of antibiotics. Consequently, antibiotics are used in excess to overcome the loss thereby bacteria under increased evolutionary pressure leading to the development of MDR and XDR pathogenic strains [6].

The current research study uses Site Identification by Ligand Competitive Saturation (SILCS) based Computer Aided Drug Design (CADD) [7–9] to identify novel inhibitors against class C β-lactamase of multi-drug resistant Enterobacteriaceae. It proposes a non-β-lactam-based β-lactamase inhibitor that might potentially inhibit broad-spectrum plasmid-encoded Class C β-lactamases. Furthermore, the strategy of combinational therapeutics has been exploited by subjecting the computationally screened compounds to experimental validation against MDR clinical isolates. The potential for application of a combinatorial therapeutic approach in to treat MDR clinical isolates is indicated through our experimental studies.

## Results

The complete protocol used in this study is shown in Fig 1.

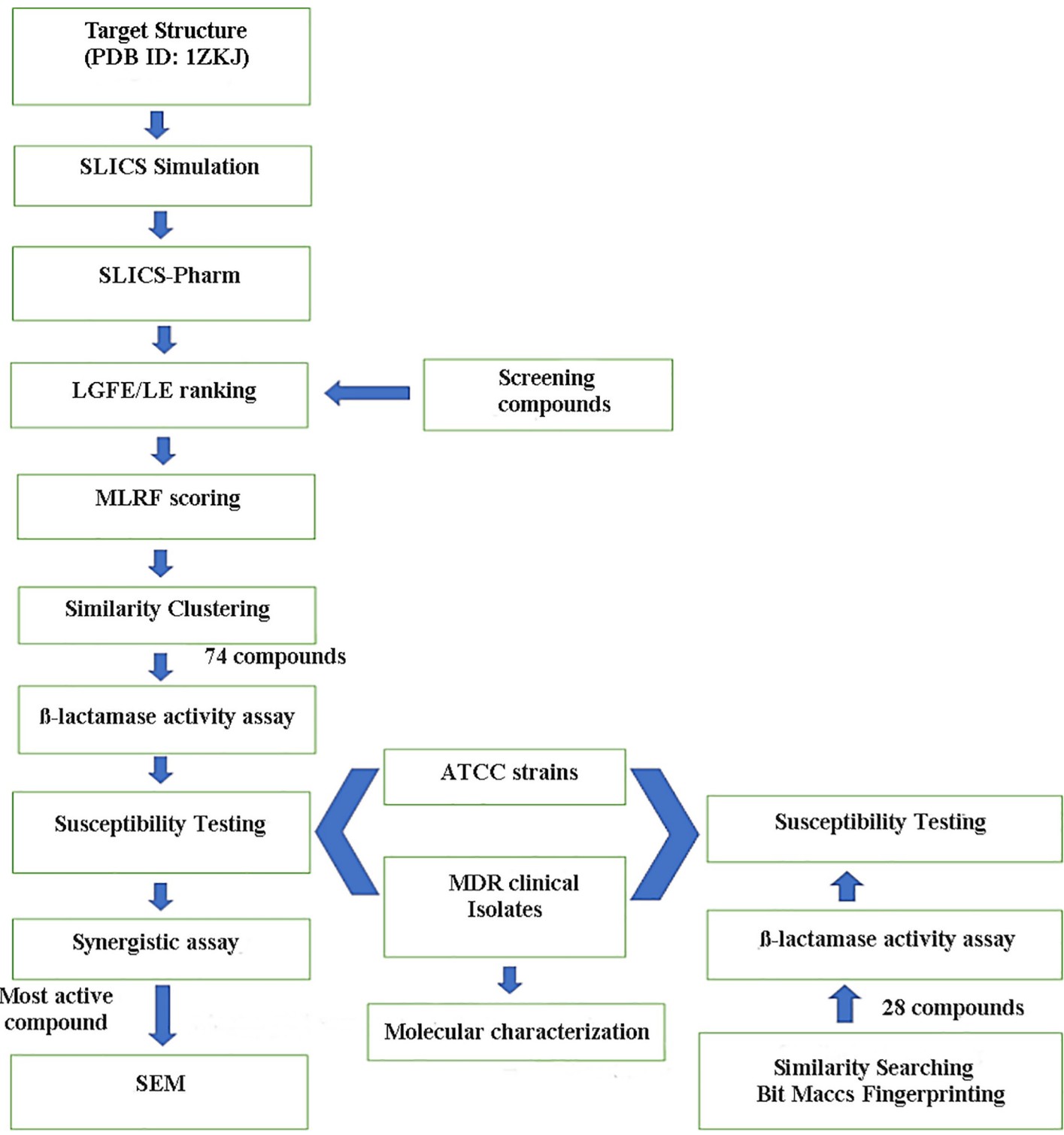

**Fig 1. The computational screening and experimental validation protocol followed to identify putative CMY-10 inhibitors for combination therapy against multi-drug resistant Enterobacteriaceae.**

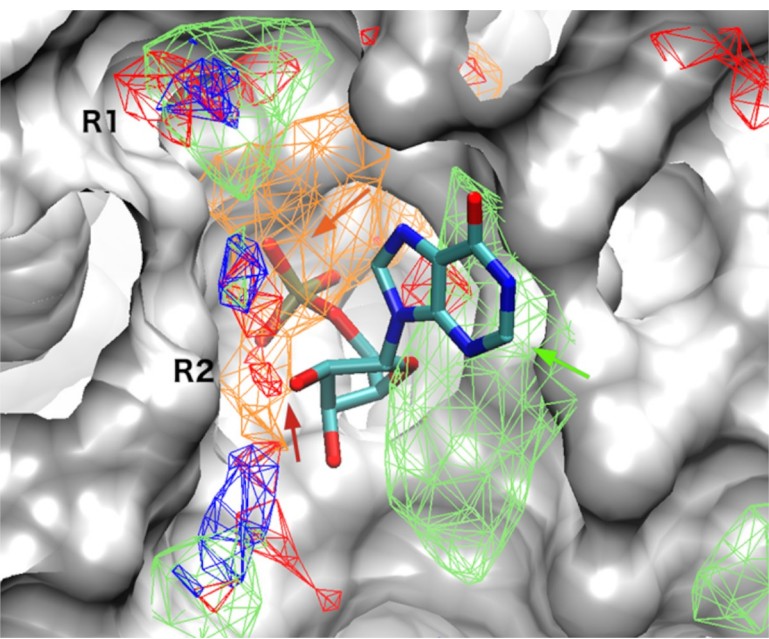

**Fig 2. Apolar (green), hydrogen bonding donor (blue) and acceptor (red) and negatively (orange) charged SILCS FragMaps overlaid on the active site of CMY-10.** The crystal binding mode of IMP in the CMY-10-IMP complex is shown. Consistencies between the crystal binding mode of IMP with FragMaps are shown by arrows.

## Computational screening

SILCS simulations were undertaken to map the functional group requirements of the active site of CMY-10 including contributions from desolvation and accounting for local protein flexibility. SILCS FragMaps at the active site of CMY-10 show apolar FragMaps at both R1 and R2 sub-sites (Fig 2). Negatively charged FragMaps are seen at the R2 site as well as some hydrogen bonding donor and acceptor FragMaps. The crystal binding mode of IMP (inosine monophosphate) from the CMY-10-IMP complex (PDB entry 5K1F) aligns well with the generated FragMaps (Fig 2). Notable is the phosphate in the negative FragMaps, the location of a hydroxyl next to an acceptor FragMap and the presence of the base in the Apolar FragMaps. The functional group binding patterns encoded in the FragMaps were further utilized to build pharmacophore features for use in Virtual Screening (VS) (S1 Fig).

The pharmacophore model that was developed for the R2 site is shown in Fig 3. Consistent with the SILCS FragMaps the pharmacophore features align well with the IMP binding mode. The model contains one hydrophobic feature (F1) corresponding to the base in IMP, an anionic feature for phosphate (F2), and hydrogen bonding acceptor feature (F3) for the sugar oxygen which forms a hydrogen bond with residue N340. In addition to features associated with the IMP binding mode, another hydrogen bonding donor feature (F4) is defined that represents functional groups interacting with the N340 carbonyl group.

To encompass both the R1 and R2 sites, the pharmacophore model in Fig 4 was generated from the FragMaps. In the figure the CMY-10 structure is aligned with the crystal structure AmpC beta-lactamase from *Escherichia coli* in complex with ceftazidime as shown in pink color (PDB ID: 1IEL). CMY-10 has high sequence similarity with AmpC, and the R1 and R2 site definitions were originally defined from studies on AmpC [5]. Accordingly, the crystal binding mode of Ceftazidime bound to AmpC is useful to inform CMY-10 R1-R2 site inhibitor design. To cover both R1 and R2 sites, two hydrophobic features (F2 and F3) were selected,

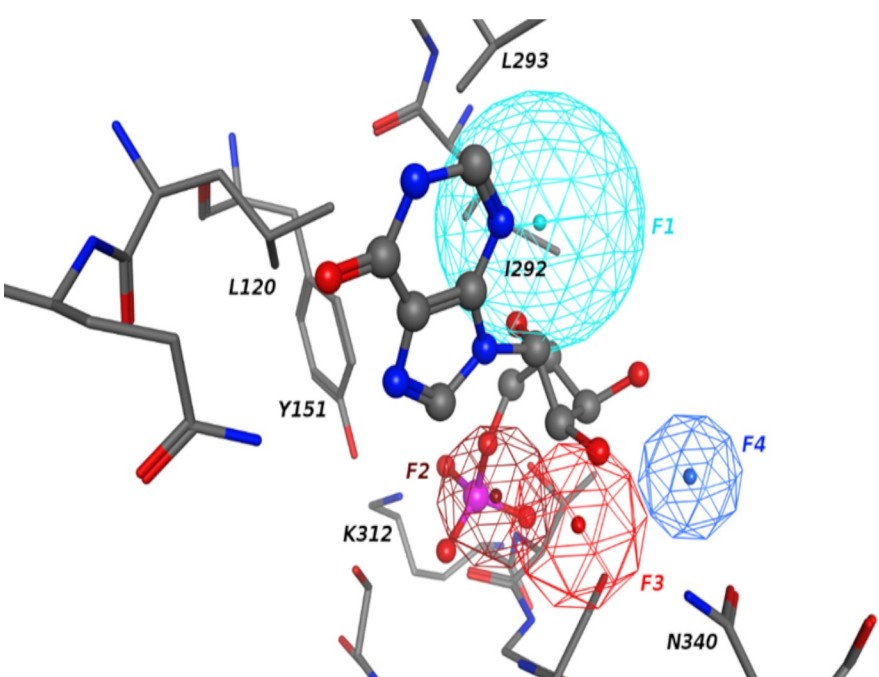

**Fig 3. Pharmacophore model for R2 site.** Hydrophobic, hydrogen bonding donor and acceptor, and anionic features are shown in cyan, blue, red and dark red colors, respectively. Crystal binding mode of IMP is also shown with surrounding protein residues labeled.

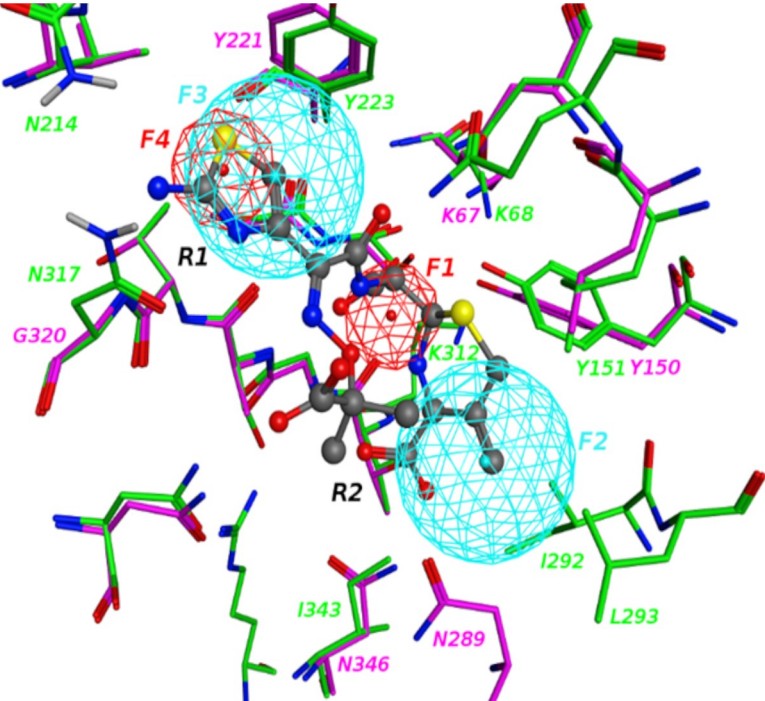

**Fig 4. Pharmacophore model for R1-R2 site.** Hydrophobic and hydrogen bonding acceptor features are shown in cyan and red colors, respectively. Crystal binding mode of Ceftazidime with AmpC is also shown with surrounding protein residues labeled. CMY-10 residues are shown in green colored carbons and AmpC residues are represented in pink colored carbons.

which align well with the crystal binding mode of the aromatic moieties in Ceftazidime, were considered. One acceptor feature (F1), which aligns well with the carbonyl group in the ligand was also included. This feature actually is at the same position of the anionic feature that was developed for the R2 site above. In this case it is set to be an acceptor feature to be more general and cover both charged and neutral acceptors as inspired by AmpC inhibitors. Though being highly similar, CMY-10 still shows some sequence difference with AmpC at the R1 site, e.g. polar residues N317 and N214 for CMY-10 while non-polar residues are at the same positions for AmpC. This motivated selection of feature F4 as an acceptor for CMY-10 at the R1 site that is not present in the AmpC binder Ceftazidime as shown.

Compounds that can match either of the two pharmacophore models are expected to have the potential to bind to the active binding site of CMY-10 at R2 sub-site in a manner similar to that of IMP or to occupy both the R1 and R2 sites, potentially with higher affinity. Such compounds would thus act as competitive inhibitors with respect to the lactam substrates of the protein.

VS were next performed using the two pharmacophores targeting over 750,000 compounds. From this pharmacophore-based VS, hit compounds were ranked using the Pharmer root mean square deviation (RMSD) score that measures the spatial similarity between the functional groups of the screened molecules and the query pharmacophore model. The top 10,000 scored compounds with the lowest RMSD were then selected from each pharmacophore search for further evaluation.

The selected 10,000 compounds from each pharmacophore screen were subsequently subjected to SILCS-MC (Site Identification by Ligand Competitive Saturation-Monte Carlo) pose refinement. Application of SILCS-MC allows the compounds to relax in the field of the Frag-Maps based on an initial orientation from the Pharmacophore screen. Ligand-grid Free Energy (LGFE) scores obtained from the SILCS-MC are then used to rank the compounds. In addition, Machine Learning Random Forest (MLRF) scoring was applied to the two sets of 10,000 compounds. From both the SILCS-MC and MLRF ranking the top 500 compounds were selected for each pharmacophore. For each pharmacophore common compounds in the SILCS-MC and MLRF top 500 ranked compounds list were then selected. This yielded 30 compounds for the R2 site and 53 compounds for the R1/R2 site. These compounds were then subjected to similarity clustering from which one compound was selected from each cluster, although most clusters only had one compound. As a result, 74 compounds were obtained for experimental testing.

## Experimental analysis

**β- lactamase activity assay.**  Beta-lactamase activity assay was performed spectrophotometrically using a chromogenic substrate Nitrocefin. The results of β-lactamase activity assay yielded compounds 1, 5, 11, 26, 36, and 47 exhibiting decreased BL-activity 0.005 u/mg, 0.3 u/mg, 0.08 u/mg, 0.0362 u/mg, 0.016 u/mg, and 0.014 u/mg, respectively, as β-lactamase inhibitors (BLA<Control) (Table 1). However, compounds 4, 5, 7, 9, 10, 12, 17, 20, 34, 69 and 71 exhibited enhanced β-lactamase activity (BLA>Control) (S1 Table, Fig 5). The LGFE distribution for top 10,000 ranked compounds from VS considering both R2 site and R1-R2 site has been shown in S2 Fig.

**Molecular identification assay.**  The plasmid contents of three MDR clinical isolates of the bacteria were analyzed by Field Inversion Gel Electrophoresis (FIGE). Three large plasmids (130 kb) were detected in all the isolates. The plasmid-encoded β-lactamase genes of three isolates were constitutively encoded AmpC β-lactamase (CMY-10) with 1,179 bp PCR product obtained using agarose gel electrophoresis that shows resistance phenotypes of three clinical isolates for the CMY-10 genes (Fig 6).

**Table 1. BL-activity and LGFE scores of lead compounds identified as inhibitors.**

| S. No. | Chembridge 1D | BL-activity u/mg | LGFE (kcal/mol) |
|---|---|---|---|
| 1 | 6096429 | 0.005 | -9.98 |
| 5 | 12728806 | 0.3 | -8.49 |
| 11 | 5524250 | 0.08 | -10.3 |
| 26 | 5241230 | 0.0362 | -9.61 |
| 36 | 77764831 | 0.016 | -9.66 |
| 47 | 7878453 | 0.014 | -9.48 |

**In vitro susceptibility testing.** Epsilometer test (E-test) was performed to quantify antimicrobial susceptibility of clinical isolates against advanced generation of macrolide and third and fourth generation antibiotics (S2 Table). Results from this assay showed that all the three clinical isolates were highly resistant except for *E. agglomerans* which was only found susceptible to fourth generation cefepime, imipenem, meropenem (S2 Table). According to the Clinical and Laboratory Standards Institute (CLSI) guidelines, the breakpoint ranges for Enterobacter species against cefixime suggest that Minimum Inhibitory Concentration (MIC) values with zone of inhibition ≤14 mm demonstrate resistant strains, between 17 mm to 14 mm are considered intermediate and ≥17 mm indicate sensitive strains [10]. The computationally selected compounds 1, 5, 11, 26, and 37 identified as β-lactamase inhibitors showed antibiotic activity against *E. agglomerans* (ATCC 31901), whereas compounds 5 and 11 were active against *E. coli (*ATCC 10536*)*, compounds 5, 11, 26, and 47 were active against *E. cloacae*

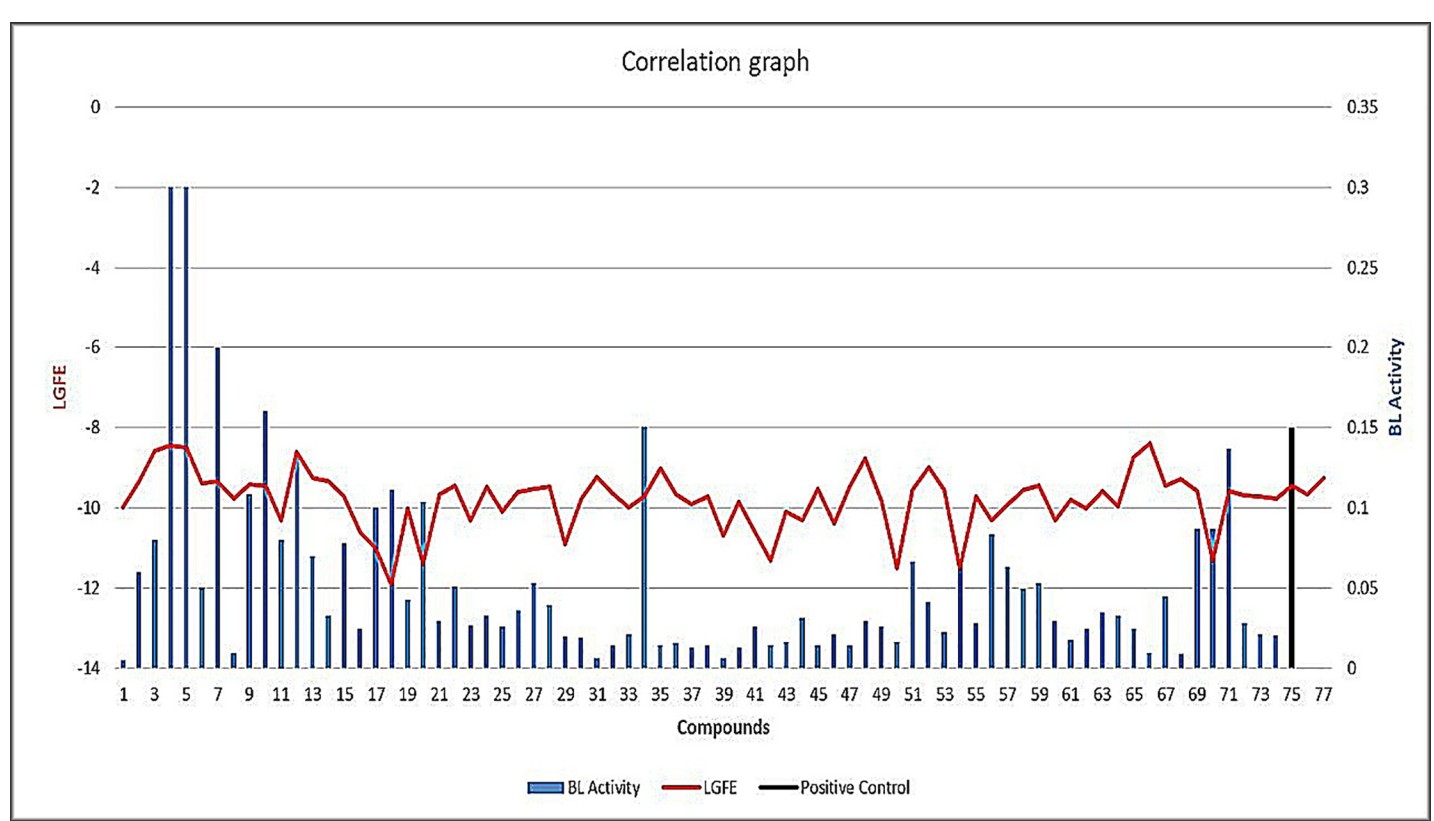

**Fig 5. BL-activity and LGFE scores of identified lead compounds.** Blue bars showing BL activity with red color indicating LGFE along with positive control in black.

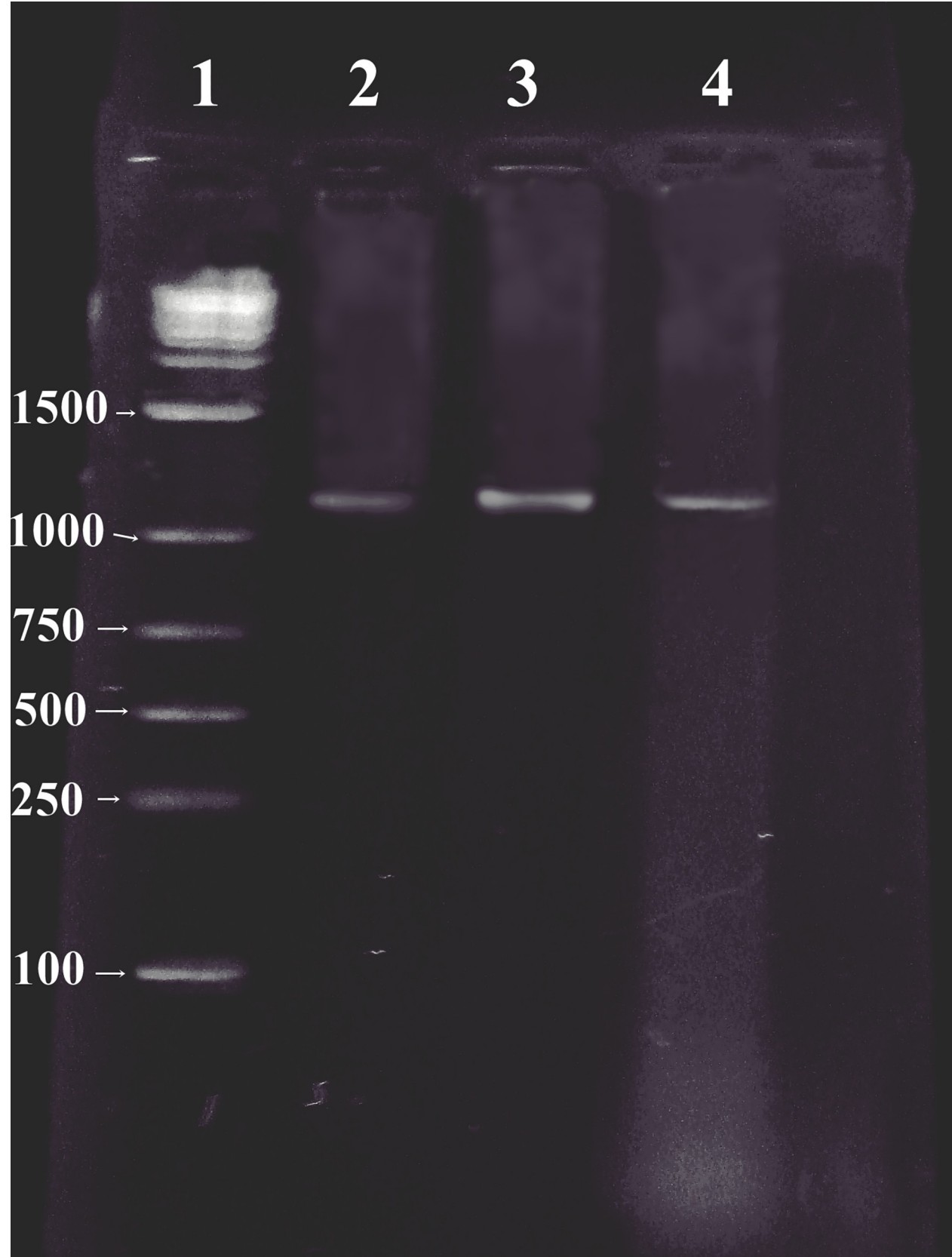

**Fig 6. Patterns of genomic plasmid from *E. cloacae* (lane 2), *E. agglomerans* (lane 3), *E. alvei* (lane 4), and lane 1 (100-bp stepwise ladder) show band patterns of ladder fragments (sizes in base pairs are indicated on the edge of the gel).**

(ATCC 13047) and compounds 11, 26, 37, and 54 showed activity against *E. alvei* (ATCC 51815) (S3 Table). However, only **11** (Chembridge ID: 5524250) showed antibiotic activity against the three multi-drug resistant clinical isolates (S3 Fig). The zone of inhibition observed for **11** and β-lactam drug cefixime (cephalosporin) used as control was <14 mm demonstrating a fair extent of resistance in clinical isolates (Table 2). MIC of compound 11 was also checked along with control imipenem and cefixime against MDR clinical isolates. The minimum zone of inhibition was measured at 4mg/ml against all the MDR clinical isolates (Table 3).

**Synergistic assay.** The synergistic assay was performed to evaluate the antibacterial activity of computationally screened compounds in combination with cefixime against MDR clinical isolates. The results of the combination study highlighted the compound **11**-cefixime combination exhibiting enhanced activity with mean zone of inhibition ≥19 mm against *E. alvei*, a clinical isolate that showed maximum resistance in susceptibility tests against computationally screened compounds and last line antibiotics. However, a breakthrough was achieved against *E. agglomerans and E. cloacae* with the compound **11**-cefixime combination showing promising activity with mean zone of inhibition in between 15 and 18 mm (Fig 7). Synergistic effect of MIC of compound 11 with minimum inhibitory concentrations of cefixime, imipenem, and amoxicillin-clavulanic acid against clinical isolates was also evaluated (Table 4). Results inferred a maximum zone of inhibition for all the three clinical strains at minimum inhibitory concentration. Herein, average mean zone of inhibition was ±20 mm against *E. cloacae*, ±19 mm against *E. agglumerans* and ±21 mm in response to *E. alvei* with minimum inhibitory level of 5ug+1ug (Cefixime: Compound 11).This response seems very effective as compare to the positive control amoxicillin + clavulanic acid. While, on other hand the combination of Imipenem and Compound 11 used in 10ug+1ug of MIC showed a potential response with ±22 mm average mean inhibition zone.

**SEM analysis.** The SEM analysis was performed to investigate the difference of morphological changes in bacterial cell walls of untreated sample and control sample in comparison with the cells exposed to the β-lactam-β-lactamase inhibitor combination (**11**-cefixime) (Fig

**Table 2. Antibacterial activity of lead compounds against β-lactamase producer clinical bacterial isolates.**

| S. No | | BACTERIAL ISOLATES (CLINICAL) | | |
|---|---|---|---|---|
| In house and Chembridge ID | | *Enterobacter Alvei* | *Enterobacter Cloacae* | *Enterobacter Agglomerans* |
| | | Zone of inhibition (mm) | | |
| | | M ± SD | M ± SD | M ± SD |
| 1 | 6096429 | 0 ± 0 | 0 ± 0 | 0 ± 0 |
| 5 | 12728806 | 0 ± 0 | 0 ± 0 | 0 ± 0 |
| 11 | 5524250 | 9.6 ±0.4 | 11.6 ±0.4 | 8.3 ±0.4 |
| 26 | 5241230 | 0 ± 0 | 0 ± 0 | 0 ± 0 |
| 36 | 77764831 | 0 ± 0 | 0 ± 0 | 0 ± 0 |
| 37 | 7989492 | 0 ± 0 | 0 ± 0 | 0 ± 0 |
| 47 | 7878453 | 0 ± 0 | 0 ± 0 | 0 ± 0 |
| 54 | 7960496 | 0 ± 0 | 0 ± 0 | 0 ± 0 |
| Control | Cefixime | 10.6 ±0.4 | 10.6 ±0.4 | 9.6 ±0.4 |

* M ± SD, Mean ± Standard Deviation, mm, millimeter.

**Table 3. Minimum inhibitory concentration of compound 11 against clinical bacterial isolates.**

| Bacterial strains | MIC test for compound 11 against clinical strains | | | | | | imipenem | cefixime |
|---|---|---|---|---|---|---|---|---|
| | MIC concentration (zone of inhibition mm) | | | | | | | |
| | 2mg/ml | 3mg/ml | 4mg/ml | 5mg/ml | 6mg/ml | 7mg/ml | 0.01mg/ml | 0.005mg/ml |
| *E. cloacae* | 0 | 0 | 7 | 10 | 12 | 14 | 8 | 7 |
| *E. agglomerans* | 0 | 0 | 7 | 10 | 12 | 15 | 8 | 7 |
| *E. alvei* | 0 | 0 | 7 | 11 | 13 | 14 | 8 | 7 |

8). Compound 11 was used in minimum inhibitory concentration in combination with cefiximie. The morphological change was apparent in cells of the pathogens treated with β-lactam-β-lactamase inhibitor combination (**11**-cefixime), in which the destruction of the cells and formations of pores were observed in the cell wall. The surface of the cell wall of untreated sample and control sample was relatively smooth compared to the cell wall of the strain treated with the β-lactam-β-lactamase inhibitor combination. The observations were consistent with experimental findings suggesting that the combined use of **11** and cefixime has enhanced therapeutic efficacy against MDR clinical isolates.

**Activity of compounds similar to compound 11.** To determine if **11** is a viable lead compound for further drug development efforts, similar compounds in the database were identified using chemical fingerprint screening [11]. Results obtained from the *β- lactamase* assay suggest that 28 compounds (1, 2, 3, 4, 5, 6, 7, 8, 10, 12, 13, 14, 15, 16, 17, 18, 20, 21, 22, 23, 24, 25 and 29) having structural similarity with the active compound exhibited *β- lactamase* inhibition potential (Fig 9, S4 Table). However, three compounds, 19, 26, and 27 showed higher

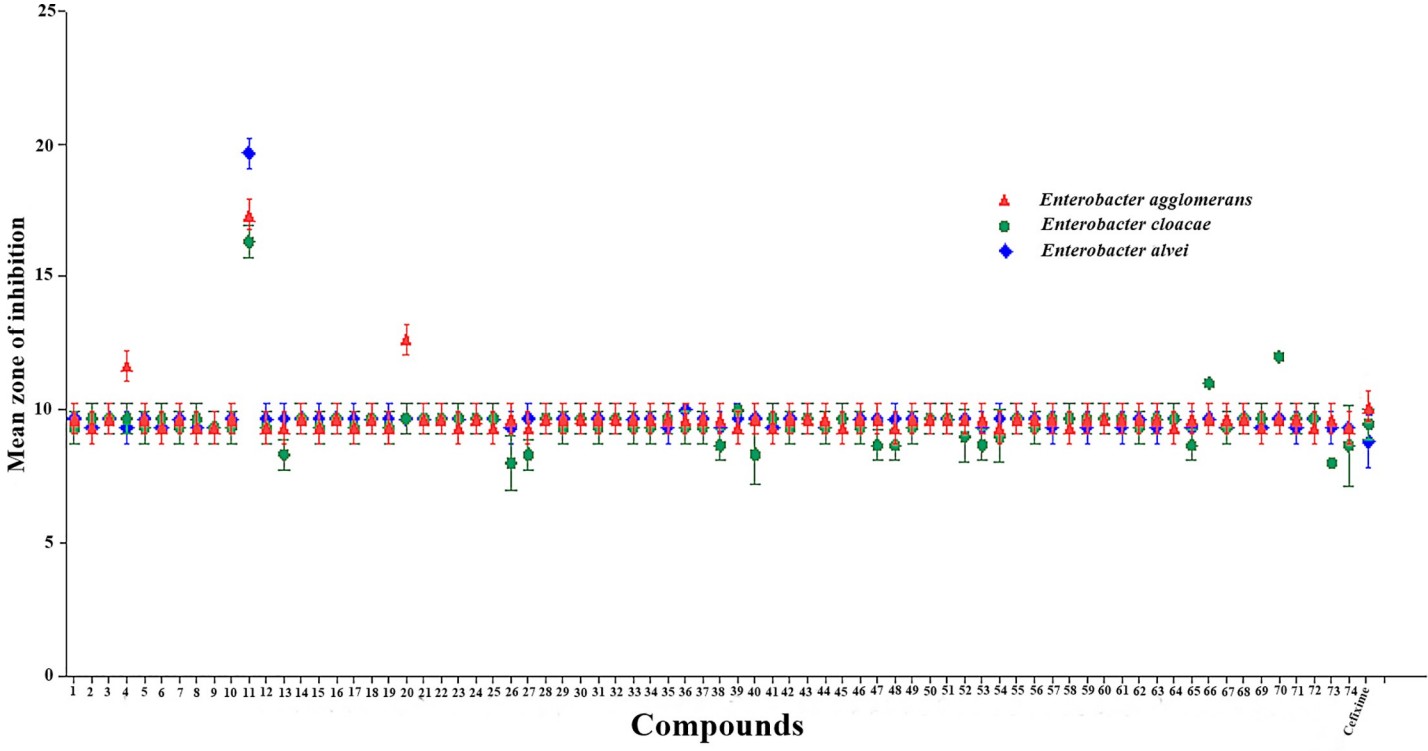

**Fig 7. Graphical representation of synergetic effect of the computationally screened compounds-cefixime with positive control.**

**Table 4. Synergistic effect of compound 11 with different concentration against clinical isolates.**

| | Synergistic effect of compound 11 against clinical antibiotic resistance strains | | | |
|---|---|---|---|---|
| | Cefixime: Compound 11 | Imipenem: Compound 11 | amoxicillin + clavulanic acid (+ve control) (20+10)µg/ml | DMSO (- ve control) |
| Bacterial strains | 5ug+1ug | 10ug+1ug | | |
| | Zone of inhibition (mm) | | | |
| E. cloacae | ±20 | ±22 | ±23 | 0 |
| E. agglumerans | ±19 | ±22 | ±20 | 0 |
| E. alvei | ±21 | ±21 | ±24 | 0 |

BL activity suggesting that these compounds might act as potential enhancers of the enzyme. Values can be inferred from S5 Table, while the activity can be visualized in Fig 9.

The similar compounds were also subjected to susceptibility testing (S6 Table). Majority of compounds exhibiting β- lactamase inhibition showed antibiotic activity against *E. agglomerans* (ATCC 31901), *E. coli (*ATCC 10536*)*, *E. cloacae* (ATCC 13047) and *E. alvei* (ATCC 51815). The exceptions were compounds 15 and 26 that were inactive against both the strains.

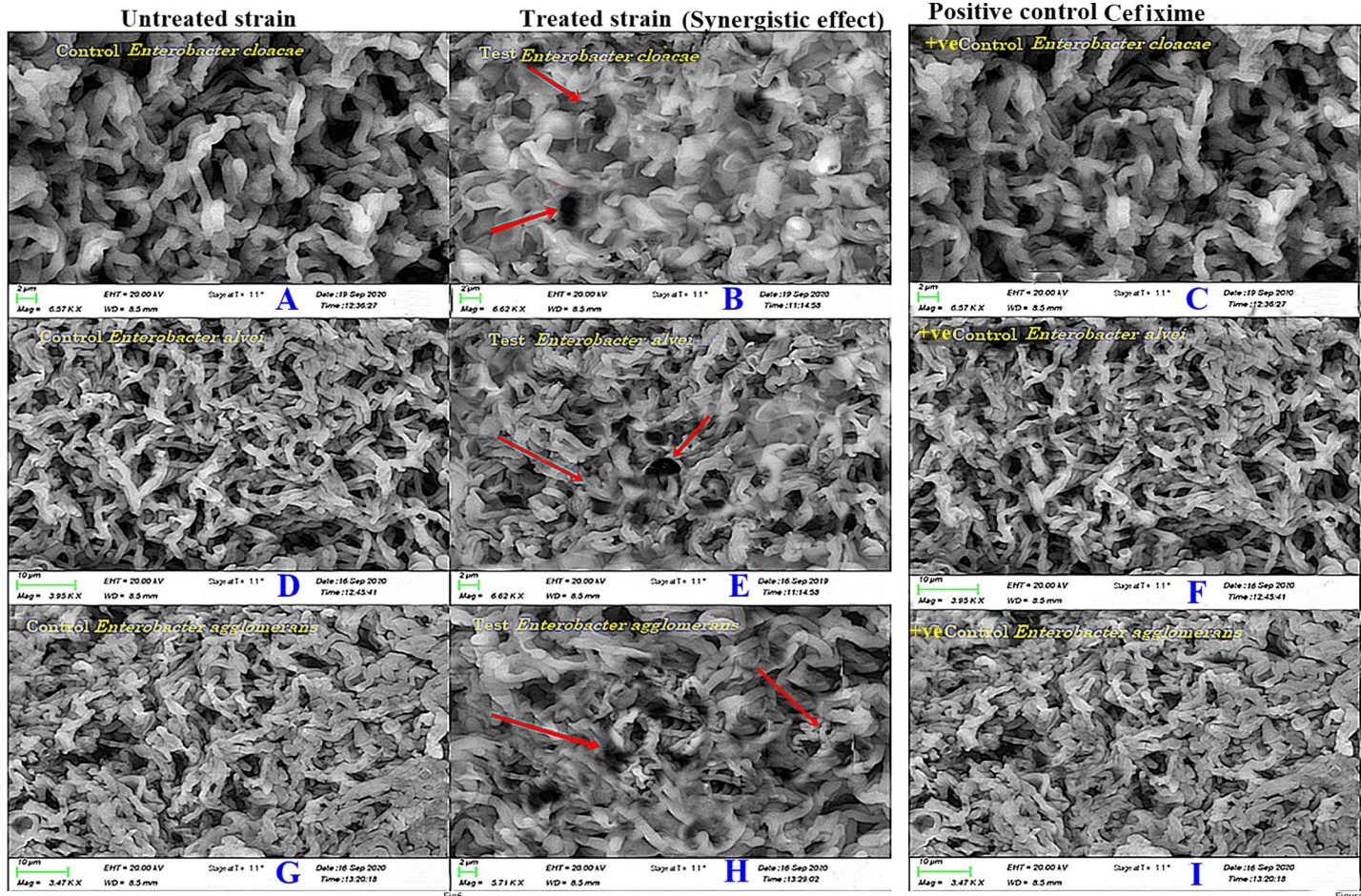

**Fig 8.** Scanning electron microscopy (SEM) analysis (A) Effects of control on *E. Cloacae* (B) Effects of lead compound 11 in combination with antibiotic, cefixime against *Enterobacter cloacae*. (C) Effects of control on *E. alvei*, (D) Effects of lead compound **11** in combination with antibiotic, cefixime against *E. alvei*. (E) Effects of control on *E. agglomerans* (F) Effects of lead compound **11** in combination with antibiotic, cefixime against *E. agglomerans*. The red arrows indicate the morphological changes exhibited on bacterial cell wall after the use of combination therapy.

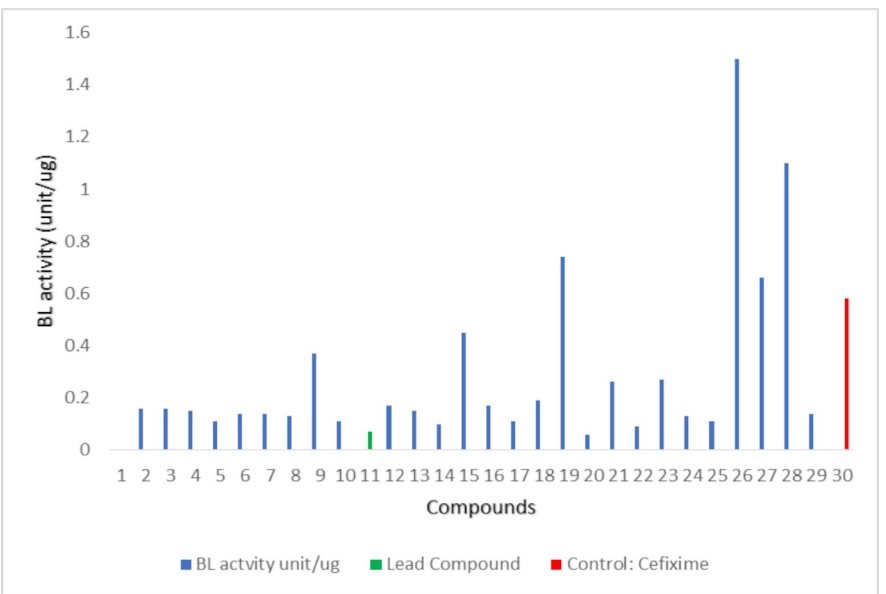

**Fig 9. BL activity of further screened compounds with negative control (Chembridge ID: 5524250) in green color and positive control in red color.**

The high zone of inhibition among these compounds was observed for compound 20 (13mm) (S6 Table). However, the results of the antibiotic assay against the MDR clinical isolates namely, *E. alvei*, *E. cloacae*, and *E. agglomerans* suggests that only compounds 5,7, **11**, 14, 20, 22, and 24 exhibited the degree of inhibition in coherence with their values obtained for respective zones of inhibition (S7 Table).

## Discussion and conclusions

ESBLs have evolved as a result of continuous use of third generation antibiotics mainly characterized by bulky oxyimino groups which were poor substrates for narrow substrate spectrum class C β-lactamases. ESBLs such as CMY-10 exhibit extended catalytic activity due to noticeable conformational changes in the active site caused by the deletion of three amino acids in the R2 loop leading to significant widening of the R2 site. The localized mutations in the R2 site of CMY-10 have played a major role in changing and extending the substrate spectrum of the enzyme contributing to its catalytic versatility. The current study has therefore employed a receptor-based CADD approach, termed SILCS, to explore the functional group binding patterns of CMY-10. This information in the form of FragMaps was then used to identify ligands with similar functional groups based on pharmacophore screening. The computational search was focused on the active site, which covers the R2-loop (residues 289–307), the flexible part of the Ω-loop (residues 212–226), α11, β11, Tyr151 loop (residues 149–152) and Gln121 loop (residues 118–128) [5]. The active site of CMY-10 can be divided into two sub-sites: R1 site (red ellipse) and R2 site (blue ellipse) (S4 Fig). The R1 side-chain of the β-lactam nucleus in β-lactam antibiotics is accommodated by the R1 site of the protein, whereas the right part of the β-lactam ring interacts with the R2 site of the receptor [5]. The R1 site is surrounded by the Ω-loop, Gln121 loop and β11, and the R2 site by Tyr151 loop, α10 in the R2-loop and α11 [5].

The active site FragMaps of CMY-10 were compared with the crystal binding mode of IMP from the CMY-10-IMP complex (Fig 2) [12]. The comparison suggests that the aromatic hypoxanthine group in IMP aligns well with apolar FragMaps (green arrow). The binding mode of

the phosphate group in IMP is captured by negatively charged FragMap (orange arrow) and a hydrogen bonding acceptor FragMap is found near the sugar hydroxyl group (red arrow) (Fig 2). These observations indicate that the SILCS FragMaps can correctly capture the binding mode of ligands interacting with CMY-10 at the active site. The information obtained from GFE FragMaps was used to generate pharmacophore models for VS of 777,605 compounds to identify potential β-lactamase inhibitors. Computational screening, including both SILCS-MC and MLRF secondary screening and similarity clustering yielded 74 non-β-lactam-based compounds which showed potential affinity to the binding pocket of the protein.

The β-lactamase activity assay results suggested that among the 74 computationally screened hits, seven compounds are β-lactamase inhibitors while eleven compounds are β-lactamase enhancers. The results of *in vitro* susceptibility testing against ATCC strains were also in accordance with β-lactamase activity assay. The β-lactamase inhibitors exhibited zone of inhibition whereas the enhancers showed no activity except compound 5 which was active against ATCC strains. The assay results revealed compounds 5, and 11 as potential inhibitors showing zones of inhibition $11.6 \pm 0.4$ and $18.3 \pm 0.4$ mm, respectively, compared with positive control cefixime exhibiting zone of inhibition of 19.6 mm against *E. coli* (ATCC 10536). However, compounds 5, 11, 26, and 47 were found active exhibiting zones of inhibition of $13.3 \pm 0.4$, $18.3 \pm 0.4$, $18.3 \pm 0.4$, and $18.3 \pm 0.4$ mm, respectively, with positive control cefixime displaying zone of inhibition of $19.3 \pm 0.4$ mm against *E. cloacae* (ATCC 13047). Compounds 1, 5, 11, 26, and 37 were active against *E. agglomerans* (ATCC 31901) showing zones of inhibition $13.3 \pm 0.4$, $8.3 \pm 0.4$, $18.3 \pm 0.4$, $11.6 \pm 0.4$, and $18.3 \pm 0.4$ mm, respectively, with positive control cefixime showing zone of inhibition of 19.3 mm. Compounds 11, 26, 37, and 54 were active against *E. alvei* (ATCC 51815) showing zones of inhibition $11.6 \pm 0.4$, $13.3 \pm 0.4$, $11.6 \pm 0.4$, and $11.6 \pm 0.4$ mm, respectively.

The predicted binding mode of compound 11 using SILCS pharmacophore screening followed by SILCS-MC is shown in S5 Fig. The compound is predicted to bind to both R1 and R2 sites. At the R2 site, 11 adopts a similar binding orientation as the crystal IMP as shown in Fig 2 with the central phenyl ring in 11 occupying the hydrophobic region of R2 site and carboxyl group reproduces the crystal binding mode of phosphate group in IMP. In addition, the chlorophenyl group in 11 occupies the hydrophobic region at R1 site. Research studies exploring structural basis of ESBLs have revealed a wider active site which decreases steric hindrance and allows the hydrolysis of third generation antibiotics [5, 13, 14]. For instance, imipenem has a long R2 side-chain which helps in forming a strained conformation of acyl-enzyme complex, preventing hydrolysis of the drug by non-extended spectrum class C β-lactamases. However, the wider R2 site of the ESBL CMY-10 allows the rapid hydrolyzation of impinem exhibiting a higher $k_{cat}$ value [5].

In a previous study by King and co-workers (2015), molecular mechanism of Avibactam-mediated β-lactamase inhibition has been discussed. Avibactam is a non-β-lactam-based β-lactamase inhibitor which has shown promising results against multidrug resistant bacterial strains such as *Pseudomonas aeruginosa* [1]. Molecular mechanism insights into the inhibitory activity of avibactam suggest that the drug forms a carbamyl linkage with nucleophilic serine in the active site of serine β-lactamases (SBLs). Similar to β-lactam-based serine β-lactamase inhibitors, the carbamyl linkage of avibactam with catalytic serine does not decompose through hydrolytic mechanism. Instead, the drug is decarbamylated by the recyclization of the diazobicyclooctane (DBO) ring reforming the inhibitor which is either released into the solution to inhibit other β-lactamases or recarbamylate in the active site of the same enzyme [1]. The possible molecular mechanism of inhibition of non-β-lactam-based inhibitors such as compound **11** in the current findings might be similar to that of avibactam-mediated reversible SBL inhibition. The identified β-lactamase inhibitors interact with the active site of β-

lactamase and might form an intermediate resistant to decomposition via hydrolytic mechanism thus affecting the catalytic activity of the enzyme.

Antibacterial activity assay was also performed on clinical isolates *E. agglomerans*, *E. cloacae*, *E. alvei* to investigate the potential of computationally screened compounds against clinical pathogens. The results of E-test confirmed the increased prevalence of MDR strains of Enterobacter species in Pakistan which are resistant to the fourth generation antibiotic including cefixime, and β-lactam antibiotics, reported as a preferred choice against ESBL producing pathogens [15]. The findings of the study by Abrar et al. also suggested an alarming increase in multi-drug resistance of *Enterobacteriaceae* in Pakistan which can have negative impact on healthcare costs, infection rates and clinical outcomes [16]. In addition to this, previous research studies have also reported the prevalence of MDR clinical isolates of *Shigella* species, *Campylobacter* species, *Salmonella typhi*, and *Neisseria gonorrhoeae* in Pakistan highlighting the need for potential interventions and development of novel therapeutics to control and reduce antimicrobial resistance [17–21]. Compound **11** which was highly active against ATCC strains also showed antibacterial activity against all three clinically isolated MDR Enterobacteriaceae strains while all the other compounds including those identified as enhancers and inhibitors remained inactive against the clinical isolates. The MIC of compound **11** exhibited mean zone of inhibition with diameters of 9.6 ± 0.4 mm, 11.6 ± 0.4 mm, and 8.3 ± 0.4 mm, while control (cefixime) having diameters of 10.6 ± 0.4 mm, 10.6 ± 0.4 mm, and 9.6 ± 0.4 mm against *E. alvei*, *E. cloacae* and *E. agglomerans*, respectively. These observations suggest that compound **11** may have high binding affinity for the PBPs of Enterobacteriaceae nearly equal to that of third generation antibiotic cefixime. This does not indicate that other computationally screened compounds identified as inhibitors do not have any activity against clinical isolates of diverse bacterial species. They may be active against other multidrug resistant strains if subjected to further experimentation. Therefore, in the future studies, the activity of the rest of the computationally screened leads should be tested against different bacterial strains to explore their antibacterial potential.

Various research studies have reported the efficacy of using a combination of β-lactam antibiotic with β-lactamase inhibitor in enhancing the potency of antibacterial agents [1, 22, 23]. In the current study, MICs were also determined using computationally screened inhibitors in combination with cefixime against MDR clinical isolates to exploit the strategy of combination therapeutics. The results of the synergistic assay were dominated by the antibacterial activity of compound **11** against MDR clinical strains. The activity of cefixime was notably improved in the presence of compound **11** against *E. agglomerans*, *E. cloacae*, and *E. alvei*. In addition, minor synergistic effect was recorded for compound **5** against *E. agglomerans* with a zone of inhibition **11** mm while the MDR clinical isolates remained resistant to the combination of cefixime with all the other computationally screened inhibitors. The effect of compound 11 along cefixime and imipenem with commercially available positive control amoxicillin + clavulanic acid was also checked. This minimum inhibitory concentration was applied according to the results obtained via microdilution disc diffusion assay. Results inferred the maximum zone of inhibition against all three clinical strains at minimum inhibitory concentration.

The morphological changes induced by using the combination of cefixime and compound **11** evaluated using SEM suggest that the use of β-lactam-β-lactamase inhibitor has negatively affected the synthesis of cell wall in all three clinical isolates. Extensive structural damage can be seen in *E. Alvei* which was higher than that observed as compared to the *E. agglomerans* and *E. cloacae* (Fig 8). Cefixime has been found ineffective against MDR clinical isolates used in this study. However, the clinical isolates have exhibited susceptibility when exposed to the combination of compound **11** and cefixime. These findings are consistent with the results of the synergistic assay suggesting that **11** might have the potential to act as β-lactam enhancer and β-lactamase inhibitor, significantly contributing to overcome antibiotic resistance.

The current study identifies computationally screened compound **11** exhibiting enhanced β-lactamase inhibitory activity in the experimental assays as a potent lead compound against MDR pathogens. The chemical structure of compound **11** (N-(2-{[(4-chlorophenyl) amino] carbonyl}-4, 6-dinitrophenyl)) contains two nitro groups and a halogen, chlorine (S3 Fig). Nitro groups are reported as best leaving groups that are smoothly displaced by nucleophiles [24]. Halogens mediate hydrogen bond donor interactions and can also act as nucleophilic acceptors widely contributing to receptor-ligand interactions. Halogens have always received significant attention in drug design due to their crucial role in enhancing selectivity and binding affinity. In addition, research studies have explicitly reported that the presence of halogen Cl facilitates interaction with hydrogen bond donor group in protein resulting in enhanced ligand binding affinity [25–27]. Research studies exploring the structure and mechanism of β-lactamase enzymes have reported the role of conserved active site serine which acts as a nucleophile to facilitate the interaction with β-lactam antibiotics and β-lactamase inhibitors [1, 5]. The presence of two nitro groups and a halogen Cl in compound 11 may have contributed to its promising activity against MDR clinical isolates in susceptibility testing and synergistic assays. In contrast to compound **11**, all the other screened compounds either do not have nitro groups or have only one nitro group in their structure. However, the combination of two nitro groups along with one halogen is only present in compound **11** which may play a crucial role in the enhanced antimicrobial and β-lactamase inhibitory activity reflected in the experimental assays. This property of having two nitro groups and one halogen can be considered as a unique property determination yielded as an outcome of the current investigation. However, extensive structural, kinetic and mutagenesis studies are required to understand the molecular details and active site features responsible of compound **11** mediated inhibition.

The potential β-lactamase inhibitor, compound **11**, proposed in the current study is not based on a β-lactam core structure which reduces its chances of getting hydrolyzed by wild and mutant β-lactamase enzymes. Non-β-lactam-based β-lactamase inhibitors are capable of escaping different pathogen resistance mechanisms mobilized against β-lactams thus reducing the chance of overexpression of β-lactamase protein [28–30]. Therefore, in order to refine the quest for potential non-β-lactam-based β-lactamase inhibitors, compound **11** was further subjected to a next round of similarity searching. As an outcome of the search, 28 compounds were found to be similar to **11**. Of these compounds, compound **20**, which keeps higher zone of inhibition against MDR clinical, also inherits two nitro groups and one halogen which may have contributed to its activity in susceptibility testing. Values obtained from β-lactamase activity assay, and susceptibility testing suggested that majority of the similarity searched compounds are also active against the enzyme (Fig 9, S5 and S6 Tables). Meanwhile, MDR clinical isolates exhibited susceptibility against compound 5, 7, **11**, 14, 20, 22, and 24 in total. The availability of multiple similar compounds with the desired antimicrobial activity indicates that **11** is a viable lead compound for further optimization and evaluation towards clinical trials. It is concurred that the use of combination therapy with non-β-lactam-based β-lactamase inhibitors can be effective against decreased susceptibility pathogens that use ESBLs as their primary resistance mechanisms. Accordingly, the compounds reported here have the potential to be effective clinical agents with the ability to circumvent the antimicrobial resistance caused by the species of Enterobacteriaceae.

## Material and methods

### Computational screening

The SILCS [9] based CADD protocol starts by using the CMY-10 protein structure to initialize SILCS simulation from which functional group requirements of the protein in terms of free

energies are obtained. The FragMaps are then used with SILCS-Pharm [31, 32] to build pharmacophore models for virtual database screening. Subsequent SILCS-MC docking [7–9] was conducted to refine the screening results, with final ranking using SILCS based ligand-grid free energy (LGFE) scores and energy scores based on a Machine Learning Random Forest (MLRF) virtual screening model [33]. Final compounds selected for the experimental assay are those common to both the LGFE and MLRF selected compounds supplemented with similarity clustering to allow for the selection of compounds of maximal diversity for testing. The details of each step of CADD protocol are addressed in the following sections.

**SILCS simulations.**    The SILCS [9] simulations involves combined Grand Canonical Monte Carlo/Molecular Dynamics (GCMC/MD) [34] simulations of the target protein immersed in an aqueous solution that contains organic solutes of different chemical classes. The solutes and water then compete for binding sites on the protein surface and in pockets in the protein during the simulation, yielding a free energy fragment competition assay from which the 3D fragment probability distributions of the solutes and water are used to define affinity patterns, termed FragMaps, encompassing a dynamic protein surface.

The current SILCS run was performed using the Grand Canonical Monte Carlo (GCMC)/ MD protocol for SILCS [34]. The target protein was solvated in a water box, the size of which is determined to have the protein extrema separated from the box edge by 12 Å on all sides. Eight representative solutes with different chemical properties (benzene, propane, acetaldehyde, methanol, formamide, imidazole, acetate, and methylammonium) were added into the system at ~0.25 M concentration, to probe the functional group requirements of the protein.

Ten such systems with different fragment positions and with the side chain chi1 dihedrals of solvent-exposed residues randomized were prepared to expedite the convergence of the simulations [8]. Each system was minimized for 5000 steps with the steepest descent (SD) algorithm [35] in the presence of periodic boundary conditions (PBC) [36] and was followed by a 250 ps MD equilibration. During SILCS simulations, weak restraints were applied on the backbone Cα carbon atoms with a force constant (k in 1/2 kδx 2) of 0.12 kcal/mol/Å$^2$ to limit large conformational changes in the protein and to prevent the rotation of the protein in the simulation box. Ten GCMC/MD simulations were run for 125 cycles where each cycle has 200,000 steps of GCMC and 1 ns of MD. The first 25 cycles included only the GCMC steps were treated as equilibration and discarded yielding a cumulative 200 million steps of GCMC and 1 microsecond of MD over the 10 simulation systems. During GCMC, solutes and water are exchanged between their gas-phase reservoirs; the excess chemical potential used to drive such exchange is varied every 3 cycles to yield an average concentration corresponding to 0.25 M of each fragment. The configuration at the end of each GCMC run was used as the starting configuration for the following MD. During MD, the Nosé−Hoover method (Hoover, 1985; Nosé, 1984) was used to maintain the temperature at 298 K and pressure was maintained at 1 bar using the Parrinello−Rahman barostat [37–39]. CHARMM36 protein force field [40], CHARMM General Force Field (CGenFF) [41, 42] and modified TIP3P water model [43] were used to describe protein fragments, and water during the simulation, respectively. GCMC was performed by an in-house code and MD was conducted using GROMACS program [42, 44].

3D probability distributions of the selected atoms from the solutes from the SILCS simulations were constructed and combined to obtain both specific and generic FragMap types as previously described [45]. Atoms from snapshots output every 10 ps from each SILCS simulation trajectory were binned into 1 Å × 1 Å × 1 Å cubic volume elements (voxels) of a grid spanning the entire system to acquire the voxel occupancy for each FragMap atom type being counted. The voxel occupancies computed in the presence of the protein were divided by the value in bulk to obtain a normalized probability. Normalized distributions were then

converted to grid-free energies (GFE) based on a Boltzmann transformation for visualization and quantitative use [45].

**SILCS-pharmacophore for CMY-10.** The SILCS-Pharmacophore (SILCS-Pharm) protocol was used to prepare pharmacophore models for virtual screenings (VS). SILCS-Pharm can generate receptor-based pharmacophore models using information from the SILCS FragMaps. This protocol includes conversion of SILCS FragMaps into pharmacophore features followed by pharmacophore hypotheses generation and ranking, with the resulting pharmacophores suitable for a range of VS tools [31, 32]. FragMaps in the active site of CMY-10 were used to find all possible pharmacophore features using the SILCS-Pharm program. Two pharmacophore models were developed for VS. One model is focused on the R2 site, which is the IMP/GMP binding site with the second pharmacophore encompassing both the R1 and R2 sites. Two four-feature pharmacophore models were developed for VS for the 1) R2 binding site alone and the 2) combined R1 and R2 sites [12]. Four-feature models were found to give the best performance as shown in previous tests [31, 32]. Compounds with patterns of functional groups that match the two pharmacophore models are expected to have the potential to bind to the active site of CMY-10 and disrupt it catalytic activity.

**VS for CMY-10.** VS was carried out to screen our in-house database of commercially available compounds using the developed two four-feature pharmacophore models. The in-house database contains 721,368 compounds (1,695,786 molecules considering different protonation states and tautomers) from the vendor Chembridge and 56,237 compounds (126,575 molecules) from the vendor Maybridge. In addition to the four features in each model, the SILCS exclusion map was also used in the model to represent the forbidden region that ligands cannot occupy. Pharmer [46] was used to carry out the pharmacophore-based VS.

**SILCS-MC for hit compounds.** Top compounds selected from the SILCS-Pharm screen were rescored using Monte Carlo (MC) sampling using SILCS FragMaps (SILCS-MC) [45]. SILCS-MC allows the docking pose of hit compounds from pharmacophore VS to relax in the field of the FragMaps using MC sampling. SILCS-MC is based on the ligand grid free energy (LGFE) score of the ligands, which is the sum of the atomic GFE contributions of the SILCS-classified non-hydrogen atoms in each ligand. Local MC sampling leads to better matching with the energetic details in the binding pocket as defined by the SILCS FragMaps. SILCS-MC was conducted in local sampling mode with 100 steps of MC followed by 1,000 simulated annealing (SA) steps to refine docking poses locally. The Metropolis criteris is based on the ligand LGFE score along with the intramolecular energy based on the CGenFF energy function along with a $1/4r$ effective dielectric constant. After SILCS-MC, each hit compound was scored based on the LGFE and on the ligand efficiency (LE), which is the LGFE divided by the number of non-hydrogen atoms. Details of the screening protocol may be found in Ustach et al. (2019).

**MLRF-score.** A machine-learning-based random-forest (MLRF) scoring scheme (RF-Score-VS) was shown to yield results similar to or better than traditional VS methods [33]. As it represents a knowledge-based alternate scoring function developed based on an approach significantly different then SILCS method, it was used to score all docked poses from SILCS-MC run with the goal of reducing false positives.

**Common-compounds selection.** Common compounds selection of the top-ranked compounds based on the individual LGFE ranked and RF-Score ranked list was performed for both the R2 and R1/R2 searches. This type of consensus scoring scheme has been shown to decrease the false positive rate [47].

**Similarity clustering.** Since all hit compounds were selected by matching the pharmacophore features, chemical structure redundancy was a concern. Similarity clustering was conducted to maximize the chemical diversity of the compounds selected for assay by clustering

hit compounds. BIT-MACCS fingerprints [48] were calculated to index all selected compounds, and Tanimoto similarity coefficients were calculated between all compound pairs and similarity clustering was performed to put chemically similar compounds into clusters. The final compounds were ranked by LGFE with cluster numbers and were further subjected to experimental validation.

## Experimental analysis

**β-lactamase activity assay.** The top selected compounds obtained from computational analysis were purchased (1mg each) from ChemBridge (https://www.chembridge.com/screening_libraries/). β-lactamase activity was detected spectrophotometrically using nitrocefin as substrate. Positive control solution along with reaction mix was prepared following the protocol provided by abcam® β-lactamase kit (ab197003). Stock solution of the selected compounds was prepared by adding 0.001 g of compound in 100 μl. The final concentration of test compounds was set to 1 μM, which were then added in 96 well microplate and screened under a multiscan spectrophotometer for absorbance at 490 nm. The samples were kept in the dark and the optical density (OD490 nm) was measured in a kinetic mode at room temperature for 30 mins. The complete absorbance upon different time intervals was then calculated i.e. initial time and the final time were then correlated. The data was further analyzed by plotting the correct absorbance values for each standard as a function of the final concentration of hydrolyzed Nitrocefin.

**Molecular identification assay.** The clinical bacterial strains *E. cloacae*, *E. alvei*, and *E. agglomerans* used in this study were obtained from the Pakistan Institute of Medical Sciences (PIMS), Islamabad, Pakistan. Molecular assay was performed to analyze whether the MDR clinical bacterial isolates are β-lactamase CMY-10 producers.

Plasmid DNA from three clinical isolates was isolated according to guidelines of Sambrook and Russell [49]. These plasmids were then separated in 1.0% agarose using a FIGE Mapper Electrophoresis System (Bio-Rad, Hercules, CA). They were purified with a Gel Extraction Kit (Genomid, Research Triangle Park, NC). The purified plasmids were utilized as a source of template DNA for PCR amplification.

The primers were obtained from the literature and utilized against the desired plasmid DNA template to identify a gene of interest. Herein, 4 mM MgCl250 pM of each primer (5´-GTAGA CCATATGCAACAACGACAATCC-3´) and C-XhoI (5´-GAATGTCTCGAGCTCTTTCTTTC AACC-3´) were used as forward and reverse primers, respectively. Amplification was carried followed by the analysis of products which was performed in 2% Seakem LE agarose (BMA, Rockland, ME). This further was followed by PCR amplifications on a thermal cycler.

**In vitro susceptibility testing.** The bacterial strains *E. agglomerans* (ATCC 31901), *E. alvei* (ATCC 51815), *E. cloacae* (ATCC 13047) and *E. coli* (ATCC 10536) were obtained from ATCC. Epsilometer test was performed against clinical isolates to check the culture sensitivity against different classes of antibiotics with minimum inhibitory dose. In order to determine a MIC with the E-test, the surface of an agar plate was swab inoculated with an adjusted bacterial suspension in the same manner as a disk diffusion test. One or more E test strips were then placed on the inoculated agar surface containing the clinical strains. The plates were incubated for 24 hrs at 37°C.

The antibacterial assays were performed to find out the inhibitory effect of the selected compounds against ATCC strains and β-lactamase producing clinical isolates. The agar disc diffusion method on Mueller Hington Agar (MHA, Oxoid, England) as defined by Lalitha [50] was applied to test the antibacterial potential of selected 74 compounds. Fresh bacterial colonies were prepared and inoculated in sterilized 1 ml of normal saline and were compared to the turbidity standard of 0.5 McFarland (1% BaCl2 and 1% H2SO4). ATCC strains and β-

lactamase producing clinical isolates were used to prepare homogenous bacterial lawn. The test inoculum was ~ 1×104 cfu/ml. Test compounds (1 μM) with positive control cefixime (cephalosporin) at the same concentration were added on disks placed on Muller Hington Agar media. The tested strains were incubated for 24 hrs at 37˚C. The minimum zone of inhibition of each compound was determined according to CLSI guidelines [10].

**Synergistic assay.** The parameters for the optimization of the antibacterial assay were set by following the protocol as discussed above. In order to check the inhibitory activity for the inhibitor molecules, the efficacy was examined in a synergistic assay against three clinical bacterial strains: *E. cloacae*, *E. alvei* and *E. agglomerans*. This technique is set to achieve the antagonistic effect of inhibitor molecules. The parameters were set to perform the assay by obtaining fresh culture of strains and preparing a bacterial lawn upon Muller Hington Agar media in petri dishes. The concentration of inhibitory molecules with cefixime drug is achieved with 1:1 μg/μl along with positive control of cefixime only in 1 μg/μl concentration. The tested strains were incubated for 24 hrs at 37˚C.

**Scanning Electron Microscopy (SEM) analysis.** The sample was prepared following the protocol used by Murtey and Ramasamy (2016) and was centrifuged to obtain solid pallet of bacterial cells of three clinical strains [51]. The pallet was washed twice and dehydrated with a gradient solvent of ethanol for 20 minutes. This examination was done to check the potency of compound 11 with cefixime at 1:1 μg/μl and control with blank sample.

**Similarity searching.** BIT-MACCS fingerprint-based similarity search was conducted for lead compound 10**11** to identify structurally similar compounds with improved biological activity. Compound 11 was searched against 5.04 million compound database and 28 compounds were selected for further assay tests (S4 Table). All previous assays mentioned above were repeated on these compounds.

## Supporting information

**S1 Table. BL-activity and LGFE scores of lead compounds identified as enhancers.**
(DOCX)

**S2 Table. Epsilometer test results for commercially available beta-lactam antibiotics against MDR clinical isolates.**
(DOCX)

**S3 Table. Antibacterial activity of lead compounds against ATCC bacterial isolates with zone of inhibition (mm).**
(DOCX)

**S4 Table. Two dimensional chemical structures of the similarity searched compounds including active compound 11.**
(DOCX)

**S5 Table. BL-activity of further screened compounds.**
(DOCX)

**S6 Table. Antibacterial activity of similarity searched compounds against ATCC bacterial isolates with zone of inhibition (mm).**
(DOCX)

**S7 Table. Antibacterial activity of similarity searched compounds against β lactamase producer's clinical bacterial isolates.**
(DOCX)

**S1 Fig. All pharmacophore features generated from SILCS FragMaps at the R1 and R2 site region.** Hydrophobic, hydrogen bond acceptor, hydrogen bond donor and negatively charged features are colored in cyan, red, blue, and dark red, respectively.
(TIF)

**S2 Fig.** LGFE distributions for top 10,000 ranked compounds from VS for both R2 model (a) and R1-R2 (b) model. The vertical dashed line indicates the LGFE cutoff for selecting the top 500 ranked compounds. The LGFE values for the seven compounds identified as inhibitors are indicated by blue arrows with their compound ID labeled. 5 hits are from the R2 model VS and 2 hits are from the R1-R2 model VS.
(TIF)

**S3 Fig. 2D chemical structure of compound 11 (Chembridge ID: 5524250).**
(TIF)

**S4 Fig.** (A) The 3D structure of CMY-10 in ribbon representation. The residues of R1 site (Ω-loop, Gln121 loop and β11) and R2 site (Tyr151 loop, α10 and α11) are in yellow and green, respectively. The binding cavity of R1 site is represented by red ellipse and R2 site is represented by blue ellipse. The nucleophile Ser65 is shown in magenta using ball and stick representation within the blue ellipse.
(TIF)

**S5 Fig. Predicted binding orientation of lead compound 11.** The binding site is shown in the same orientation as in the Fig 2. FragMaps are shown at GFE cutoff -1.0 kcal/mol for apolar (green), hydrogen bonding donor (blue) and acceptor (red) maps and at -1.5 kcal/mol for negatively (orange) charged map.
(TIF)

## Acknowledgments

We thank Pakistan–United States Science and Technology Cooperation Program and Higher Education Commission (HEC) Pakistan.

## Author Contributions

**Investigation:** Faisal Ahmad, Wenbo Yu, Alexander D. MacKerell, Jr., Syed Sikander Azam.

**Methodology:** Nousheen Parvaiz, Faisal Ahmad, Wenbo Yu, Alexander D. MacKerell, Jr.

**Project administration:** Alexander D. MacKerell, Jr., Syed Sikander Azam.

**Supervision:** Alexander D. MacKerell, Jr., Syed Sikander Azam.

**Writing – original draft:** Nousheen Parvaiz, Faisal Ahmad, Wenbo Yu, Alexander D. MacKerell, Jr., Syed Sikander Azam.

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
