## [Decision Letter · Decision Letter 0]

11 Sep 2020

PONE-D-20-14041

Discovery of beta-lactamase CMY-10 inhibitors for combination therapy against multi-drug resistant Enterobacteriaceae

PLOS ONE

Dear Dr. Syed Sikander Azam,

Thank you for submitting your manuscript to PLOS ONE. After careful consideration, we feel that it has merit but does not fully meet PLOS ONE’s publication criteria as it currently stands. Therefore, we invite you to submit a revised version of the manuscript that addresses the points raised during the review process.

We look forward to receiving your revised manuscript.

Kind regards,

Massimiliano Galdiero, M.D., Ph.D.

Academic Editor

PLOS ONE

Journal Requirements:

2. Thank you for including your competing interests statement; "The authors have declared that no competing interests exist. ADM is co-founder and CSO of SilcsBio LLC."

3. Please include a copy of Table 6 which you refer to in your text on page 25

Reviewers' comments:

Reviewer's Responses to Questions

**Comments to the Author**

1. Is the manuscript technically sound, and do the data support the conclusions?

Reviewer #1: Yes

Reviewer #2: Partly

Reviewer #3: Partly

2. Has the statistical analysis been performed appropriately and rigorously? 

Reviewer #1: Yes

Reviewer #2: Yes

Reviewer #3: N/A

3. Have the authors made all data underlying the findings in their manuscript fully available?

Reviewer #1: No

Reviewer #2: Yes

Reviewer #3: Yes

4. Is the manuscript presented in an intelligible fashion and written in standard English?

Reviewer #1: Yes

Reviewer #2: Yes

Reviewer #3: Yes

5. Review Comments to the Author

Reviewer #1: This is an interesting paper highlighting an innovative approach at antimicrobial resistance of Enterobacteriaceae spp.

The paper is well written and detailed.

However, I have some questions about article:

Authors should explain a potential action mechanism of compound 11.

Can author clarify the correlation between lab tests and potential clinical applications?

Reviewer #2: In the present study, the Authors describe an experimental model to discover beta-lactamase inhibitors, focusing on the characterization of the antimicrobial properties of the identified compounds. The work is very interesting but in my opinion some aspects need to be improved to make the study more clear and comprehensible to the reader.

Introduction

In the nomenclature of bacterial species of Enterobacter it is sufficient to write the Genus in full for the first time, then the dotted Genus can be used (for example E. agglomerans);

In the last paragraph of the introduction, write in full "site identification by ligand competitive saturation" and put the acronym SILCS in brackets, as it is mentioned for the first time in this section.

Results

When a bacterial species is named for the first time, the name must be written in full (Escherichia coli);

To make the comprehension clearer for a non-expert reader, Authors should write in full and clarify the meaning of the acronyms "VS", "MC", "LGFE" and "MLRF";

Experimental analysis

It would be clearer if the Authors already specified in the section Results that the Beta-lactamase activity assay was performed spectrophotometrically using a chromogenic substrate such as Nitrocefin;

The legend of Figure 4 should be described more clearly, it is not clear;

Add the sentence "in vitro" in the title "Susceptibility testing";

In vitro susceptibility tests were performed on clinical isolates of Enterobacter spp. the authors should have used ATCC reference strains of the same Genus. This aspect needs to be clarified, the sensitivity towards the molecule could be different between bacterial strains of different genera;

The first time the acronym MIC is used, the meaning must be written in full;

It would be interesting and would been strengthened the data if the in vitro susceptibility tests were also carried out in broth dilution in order to verify a bacteriostatic or bactericidal effect of the compounds;

In the SEM analysis paragraph the authors should clarify the observations, the paragraph is not clear, should be rewritten. Furthermore, the resolution of Figure 6 is very poor, it is not possible to appreciate the “extensive structural damage” reported.

In the legend of Figure 6, the nomenclature of the bacterial species must be corrected, the species must be always written in lower case;

Also in the paragraph "Activity of compounds similar to compound 11" the authors should clarify that the reference strains used do not belong to the same genus as the clinical isolates, and the test should be carried out with an Enterobacter reference strain. In my opinion is wrong to use reference bacterial strains belonging to different genera, indeed the behavior of the compounds selected between ATCC strains and clinical isolates is not comparable (see compound 20).

From the main text it would seem that it has not been verified whether the analyzed bacterial strains are β-lactamase CMY-10 producers. This should be clearly demonstrated for example with a molecular assay;

The resolution of Figures 4, 5, 6 and 7 is too poor, these are not clear. In Figure 5 the written are not legible;

In my opinion, to make the tables (Table 2; S4; S6 and S7) more streamlined, the Authors could directly report the average value without reporting the values of the individual batches, and Figure 8 should be moved as the first figure, to make the experimental design clearer.

Discussion

In the discussion paragraph the Authors indicate "disc diffusion assay", but in the Results section the Authors indicate "epsilometric test". However, the diameters of the inhibition zones are commented in the section Results. What kind of test was done? This aspect is confusing and should be clarified;

In my opinion, the results of the agar diffusion tests should be commented in the Results section, making the discussion more streamlined.

Material and Methods

The bacterial suspension must be indicated with “CFU/ml” and not “CFU/spot”. Replace the sentence;

E-test strips containing the tested compounds were produced by the authors? This aspect should be clarified and the specific MIC values obtained should be reported in the Results;

In the Synergistic assays the molecules were only tested in a 1: 1 ratio? Different ratio should also be checked.

Table S3, which reports a partial antibiogram of clinical strains, should be completed and improved, so it is not clear.

Reviewer #3: The article “Discovery of beta-lactamase CMY-10 inhibitors for combination therapy against multidrug resistant Enterobacteriaceae” by Parvaiz and coworkers deals with the identification of a novel β-lactamase inhibitors. The study is of interest because is crucial to support investigation to counter the antimicrobial resistance development in pathogens. However, the study is overall preliminary and lacks some crucial experimental work to drive proper conclusion. The title mentions beta-lactamase CMY-10 but the paper lacks an experimental demonstration that substance 11 specifically inhibits this enzyme. How do the authors explain this?

ABSTRACT

The abstract needs to be rewritten to reflect the aim, methods, results, and conclusions. For example, the sentence “Of these, one compound shows promising activity in β-lactamase activity assay, susceptibility testing against ATCC strains and MDR clinical isolates, with synergistic assay indicating its potential as a β-lactam enhancer and β-lactamase inhibitor” is not clear: what are the strains analyzed? what do the authors mean by “susceptibility testing”? Please rephrase.

P. 8, Line 14: Please change “11 compounds” with “Eleven compounds”.

P. 8, Line 18-20: “Structural similarity search against the active compound yielded 28 more compounds, the majority of which also showed β-lactamase inhibition potential and antibacterial activity”. This sentence is not clear, please rephrase.

AUTHOR SUMMARY

P. 9, Line 10-12: , please rephrase the sentence “Compounds screened as potential inhibitors of CMY-10 in the current study have the potential to be used in combination therapy as non-β-lactam-based β-lactamase inhibitors against MDR clinical isolates that have been found resistant against last-line antibiotics”, because the same sentence is already written in the abstract. Some sentences are repetitions of what previously said.

INTRODUCTION

Please, explain briefly the classification of �-lactamases and the reason for the choice of identify new inhibitors only against class C �-lactamase.

RESULTS

P.12: IMP should be spelled out

P.13: The names of the bacterial species in the all text must be written in italics (E. coli)

P.14, line 20: The experimental strategy used in β-lactamase activity assay is only briefly explained in the text. Clarification on this point may help. Which strains were used in this assay?

P.15, line 1: what is the Control?

P.15, line 2: The order of Supplementary Table is wrong…. Table S2 appears first in the text while Table S1 is at the end of the paper in the materials and methods section

P.15, line 8-10: “Epsilometer test (E-test) was performed to quantify antimicrobial susceptibility of clinical isolates against advanced generation of macrolide and third and fourth generation antibiotics”: Please indicate antibiotics or refer to the supplementary table

P.15, line 12: Table S3 is not clear and incomplete, the header is wrong, some of the data refer to Epsiolmeter test results (the bottom), the other to the diffusion test…. Please explain. Why are imipenem and meropenem antibiotics not also tested?

P.16, line 1-2: Why do the authors test the substances against ATCC reference strains of S. aureus and E. coli? why don't they test the substances against a reference ATCC strain of Enterobacter? Please, perform these experiments

P16, line 3: “three multi-drug resistant clinical isolates”: only 3 isolates of Enterobacter spp were tested. Please, increase the number of clinical isolates tested.

P.16, line 6: Table 2: indicate the method used in the antibacterial activity assays. What are the 3 batches? three replicated experiments? if yes, please report only the average with DS in the table (this also applies to the other tables in supplementary).

What is the MIC of compound 11? It is preferable to determine MIC of this compound by using the broth microdilution method. Please, perform these experiments.

P.17, line 1: the synergy assay should be performed using the microbroth checkerboard method. The authors should perform this experiment for at least compound 11 and discuss the results.

P.17, line 2: Why have you chosen this antimicrobial (cefixime)? Please explain.

P.17, line 12: Fig. 6: the resolution of the figure is too low and this does not allow to verify the data. What is the control of experiment? the untreated strains? the images of the strains treated only with cefixime are missing.

P.18, line 3: “Results obtained from the β- lactamase assay suggest that”…. Please indicate the total number of compounds similar to compound 11 found (28)

P.18, line 4: “having structural similarity”: is the chemical structure of the compounds shown in Table S1? if yes, please refer to the above table.

DISCUSSION

The Discussion section is a bit long and it would also benefit from some condensation.

MATERIAL AND METHODS

P.24, line 20: In my opinion, the sentence “The complete protocol used in this study is shown in Fig 8“ should be moved to the beginning of the results, in this way the experimental procedure is immediately clear to the reader of the manuscript.

P.29, line 4: The method described for the β-lactamase activity assay is not clear. The authors should describe in detail the method used. Which sample is used in the assay? bacteria? which bacterial strains? what are the conditions used to grow bacteria?

P.29, line 15: describe punctually the protocol

P.29, line 16: did the authors characterize (at the genomic level) the clinical strains of Enterobacter used? do these strains contain the gene that encodes for CMY-10? why did the authors not test Enterobacter ATCC reference strains? Please, perform these experiments.

P.29, line 22: Change the sentence: “The test was run for 24 hrs at 37 °C” with the sentence: “The plates were incubated for 24 hrs at 37 °C”

P.31, line 4: Table S1?????? The order of Supplementary Table is wrong.

6. PLOS authors have the option to publish the peer review history of their article (what does this mean?). If published, this will include your full peer review and any attached files.

Reviewer #1: No

Reviewer #2: No

Reviewer #3: No

---

## [Author Response · Author response to Decision Letter 0]

2 Dec 2020

Response to Reviewer Comments

We thank the Referees for spending their time and interest in our work. We have checked all the comments and have made necessary changes accordingly. 

-Reviewer 1

1) Authors should explain a potential action mechanism of compound 11.

Author Response: Compound 11 is a non-β-lactam-based inhibitor. The possible action mechanism has been discussed on P. 17 from Line 333-337.

2) Can author clarify the correlation between lab tests and potential clinical applications?

Author Response: As an evaluation of the potential clinical utility of the inhibitor identified in our study, we evaluated if known antibiotics in the presence of this inhibitor proved to be more effective based on maximum zone of inhibition as compared to the antibiotics given alone in resistant clinical isolates. The data from these lab experiments indicates the potential clinical utility of the identified inhibitor in combination therapy for the treatment of MDR pathogens. This is discussed on P.20-21 Line 415-421 of the revised manuscript.

-Reviewer 2

1) In the nomenclature of bacterial species of Enterobacter, it is sufficient to write the Genus in full for the first time, then the dotted Genus can be used (for example E. agglomerans);

Author Response: The Genus has been written in full for the first time on P.4 line 63-64. 

2) In the last paragraph of the introduction, write in full "site identification by ligand competitive saturation" and put the acronym SILCS in brackets, as it is mentioned for the first time in this section.

Author Response: It has been corrected in the revised manuscript. 

3) When a bacterial species is named for the first time, the name must be written in full (Escherichia coli)

Author Response: It has been modified in the revised manuscript. 

4) To make the comprehension clearer for a non-expert reader, Authors should write in full and clarify the meaning of the acronyms "VS", "MC", "LGFE" and "MLRF"

Author Response: The acronyms have been written in full to clarify the meaning in the revised form of the manuscript.

5) Experimental analysis

It would be clearer if the Authors already specified in the section Results that the Beta-lactamase activity assay was performed spectrophotometrically using a chromogenic substrate such as Nitrocefin;

Author Response: Thank you for your valuable comment. The complete sentence has been added to the results section for clarity on P. 9 line 164-170 of the revised manuscript.

6) The legend of Figure 4 should be described more clearly; it is not clear;

Author Response: Now legend of Figure 4 has been modified for clarity.

7) Add the sentence "in vitro" in the title "Susceptibility testing";

Author Response: ‘In vitro’ has been added to the title ‘Susceptibility testing’ in overall test of the revised manuscript.

8) In vitro susceptibility tests were performed on clinical isolates of Enterobacter spp. the authors should have used ATCC reference strains of the same Genus. This aspect needs to be clarified, the sensitivity towards the molecule could be different between bacterial strains of different genera

Author Response: Thank you for your valuable comment. In vitro susceptibility testing was performed again using only the ATCC reference strains of Enterobacter spp. The results section and analysis has been revised accordingly on P.10 line 194-196 and P.16 line 308-315.

9) The first time the acronym MIC is used, the meaning must be written in full;

Author Response: The acronym MIC has been written in full, the first time it is used to clarify its meaning on P.10 line 192-193 in the revised manuscript. 

10) It would be interesting and would been strengthened the data if the in vitro susceptibility tests were also carried out in broth dilution in order to verify a bacteriostatic or bactericidal effect of the compounds

Author Response: The disc diffusion assay was performed again specifically for compound 11 against MDR clinical isolates with different dosage concentration to check the Minimum inhibitory effect of compound 11 with respect to broad spectrum imipenem and cefixime antibiotics. The results have been added on P.10 line 197-203 in the revised manuscript.

11) In the SEM analysis paragraph, the authors should clarify the observations, the paragraph is not clear, should be rewritten. Furthermore, the resolution of Figure 6 is very poor, it is not possible to appreciate the “extensive structural damage” reported.

Author Response: Thank you for your valuable comment. The written SEM analysis paragraph has been improved. Furthermore, the resolution of Figure 6 has also been enhanced. 

12) In the legend of Figure 6, the nomenclature of the bacterial species must be corrected, the species must be always written in lower case

Author Response: The nomenclature of bacterial species has been corrected in the legend of Figure 7.

13) In the paragraph "Activity of compounds similar to compound 11" the authors should clarify that the reference strains used do not belong to the same genus as the clinical isolates, and the test should be carried out with an Enterobacter reference strain. In my opinion is wrong to use reference bacterial strains belonging to different genera, indeed the behavior of the compounds selected between ATCC strains and clinical isolates is not comparable (see compound 20).

Author Response: The experiments have been rerun with the ATCC reference strains of the same genus as the clinical isolates and results are added in the line 264-271 on page 14 of the revised manuscript.

14) From the main text it would seem that it has not been verified whether the analyzed bacterial strains are β-lactamase CMY-10 producers. This should be clearly demonstrated for example with a molecular assay

Author Response: Molecular assay was performed which verified that clinical bacterial isolates are β-lactamase CMY-10 producers. The methods and results of this genome characterization experiment have been added in the corresponding sections on P.9 line 176-180 and P.26 line 527-541. 

15) The resolution of Figures 4, 5, 6 and 7 is too poor, these are not clear. In Figure 5 the written are not legible

Author Response: The resolution of the figures has been improved and figure 5 has been modified in the revised manuscript. 

16) In my opinion, to make the tables (Table 2; S4; S6 and S7) more streamlined, the Authors could directly report the average value without reporting the values of the individual batches

Author Response: Thank you for your comment. The tables have been revised accordingly in revisions.

17) Figure 8 should be moved as the first figure, to make the experimental design clearer.

Author Response: Thank you for your valuable comment. The figure 8 has been moved as first figure in the revised manuscript. 

18) In the discussion paragraph the Authors indicate "disc diffusion assay", but in the Results section the Authors indicate "epsilometric test". However, the diameters of the inhibition zones are commented in the section Results. What kind of test was done? This aspect is confusing and should be clarified

Author Response: Thank you for highlighting this point. The term ‘disc diffusion assay’ has been replaced with ‘in vitro susceptibility testing’ for clarification. 

19) The results of the agar diffusion tests should be commented in the Results section, making the discussion more streamlined.

Author Response: The results have been discussed on P.15-16 line 304-313 of the revised manuscript.

20) The bacterial suspension must be indicated with “CFU/ml” and not “CFU/spot”. Replace the sentence

Author Response: Thank you for highlighting the mistake. It has been corrected. 

21) E-test strips containing the tested compounds were produced by the authors? This aspect should be clarified and the specific MIC values obtained should be reported in the Results

Author Response: E-test was performed for evaluating the antibiotic resistance of the clinical isolates used in the study against commercially available antibiotics. E-test was not performed for the compounds reported in the study. On other hand we have also confirmed the MIC value for the compound to be reported against clinical isolates in the revised manuscript.

22) In the Synergistic assays the molecules were only tested in a 1: 1 ratio? Different ratio should also be checked.

Author Response: Thank you for your valuable comment. The synergistic assay has been run again at minimum inhibitory concentration level against all the three clinical strains. We have checked the effect of compound 11 along Cefixime and Imipenem with commercially available positive control amoxicillin + clavulanic acid P.12 line 225-231. This minimum inhibitory concentration was applied according the results obtained via micro dilution disc diffusion assay (P.18 line 373) in the revised manuscript. 

23) Table S3, which reports a partial antibiogram of clinical strains, should be completed and improved, so it is not clear.

Author Response: The table has been revised accordingly.

-Reviewer 3

1) The title mentions beta-lactamase CMY-10 but the paper lacks an experimental demonstration that substance 11 specifically inhibits this enzyme. How do the authors explain this?

Author Response: Thank you for your comment. Yes, we have done an experimental demonstration of CMY-10 target. Molecular identification assay was performed to check if the clinical isolates are CMY-10 producers or not. The results of the assay suggested that clinical isolates selected in the study are CMY-10 producers, therefore, further to address revisions experimental studies have been carried on selectel clinical isolates and inhibition results have been reported accordingly. 

2) The abstract needs to be rewritten to reflect the aim, methods, results, and conclusions. For example, the sentence “Of these, one compound shows promising activity in β-lactamase activity assay, susceptibility testing against ATCC strains and MDR clinical isolates, with synergistic assay indicating its potential as a β-lactam enhancer and β-lactamase inhibitor” is not clear: what are the strains analyzed? what do the authors mean by “susceptibility testing”? Please rephrase.

Author Response: The abstract is divided into sections to reflect aim (P.2 line 24-26), methods (P.2 line 26-33), results (P.2 line 33-39), and conclusions (P.2 line 39-41). The sentence has been rephrased and the names of the ATCC and MDR clinical isolates have been added. 

3) P. 8, Line 14: Please change “11 compounds” with “Eleven compounds”.

Author Response: The correction has been done on P.2 line 33. 

4) P. 8, Line 18-20: “Structural similarity search against the active compound yielded 28 more compounds, the majority of which also showed β-lactamase inhibition potential and antibacterial activity”. This sentence is not clear, please rephrase.

Author Response: The sentence has been rephrased for clarity. 

5) P. 9, Line 10-12: , please rephrase the sentence “Compounds screened as potential inhibitors of CMY-10 in the current study have the potential to be used in combination therapy as non-β-lactam-based β-lactamase inhibitors against MDR clinical isolates that have been found resistant against last-line antibiotics”, because the same sentence is already written in the abstract. Some sentences are repetitions of what previously said.

Author Response: The sentence has been rephrased in revised manuscript.

6) Please, explain briefly the classification of β-lactamases and the reason for the choice of identify new inhibitors only against class C β-lactamase.

Author Response: The classification has been explained briefly on Pg. 4 lines 68-72 

7) P.12: IMP should be spelled out

Author Response: Thank you for highlighting this mistake. IMP has been spelled out the first time it has been written on P. 6 line 105

8) P.13: The names of the bacterial species in the all text must be written in italics (E. coli)

Author Response: Thank you for highlighting this mistake. It has been corrected accordingly.

9) P.14, line 20: The experimental strategy used in β-lactamase activity assay is only briefly explained in the text. Clarification on this point may help. Which strains were used in this assay?

Author Response: We have used β-lactamase enzyme instead of β-lactamase producer strains in inhibition assay. 

10) P.15, line 1: what is the Control?

Author Response: The control used is cefixime and have been added on P.18 line 352.

11) P.15, line 2: The order of Supplementary Table is wrong…. Table S2 appears first in the text while Table S1 is at the end of the paper in the materials and methods section

Author Response: Thank you highlighting this mistake. The order of all the supplementary tables has been revised. 

12) P.15, line 8-10: “Epsilometer test (E-test) was performed to quantify antimicrobial susceptibility of clinical isolates against advanced generation of macrolide and third and fourth generation antibiotics”: Please indicate antibiotics or refer to the supplementary table

Author Response: The supplementary table has been referred in the text on P.10 line 190-191 by adding fourth generation antibiotics imipenem, meropenem in the revised manuscript. 

13) P.15, line 12: Table S3 is not clear and incomplete, the header is wrong, some of the data refer to Epsiolmeter test results (the bottom), the other to the diffusion test…. Please explain. Why are imipenem and meropenem antibiotics not also tested?

Author Response: Table S3 has been revised accordingly. Antibiotics imipenem and meropenem have been added in the experimental testing while adding revisions.

14) P.16, line 1-2: Why do the authors test the substances against ATCC reference strains of S. aureus and E. coli? why don't they test the substances against a reference ATCC strain of Enterobacter? Please, perform these experiments

Author Response: The experiments have been re-run to add reference ATCC strains of Enterobacter while other are excluded in instructed.

15) P16, line 3: “three multi-drug resistant clinical isolates”: only 3 isolates of Enterobacter spp were tested. Please, increase the number of clinical isolates tested.

Author Response: The data has been collected from Pakistan Hospital. Therefore, the available three clinical isolates of Enterobacter that are prevalent in Pakistan were selected in the study.

16) P.16, line 6: Table 2: indicate the method used in the antibacterial activity assays. What are the 3 batches? three replicated experiments? if yes, please report only the average with DS in the table (this also applies to the other tables in supplementary).

Author Response: Yes, these were 3 batches indicating three replicated experiments. The table has been modified for further clarification in the revised manuscript.

17) What is the MIC of compound 11? It is preferable to determine MIC of this compound by using the broth microdilution method. Please, perform these experiments.

Author Response: MIC was evaluated using disc diffusion assay. Please see table 3 and 4 for reference. 

18) P.17, line 1: the synergy assay should be performed using the microbroth checkerboard method. The authors should perform this experiment for at least compound 11 and discuss the results.

Author Response: Synergistic effect of minimum inhibitory concentration of compound 11 with minimum inhibitory concentrations of cefixime, imipenem, and positive control amoxicillin-clavulanic acid against clinical isolates was also evaluated in the revised manuscript in page 12 line 225-231.

19) P.17, line 2: Why have you chosen this antimicrobial (cefixime)? Please explain.

Author Response: Imipenem has also been added now. Cefixime has been selected because it is third generation antibiotic whereas Imipenem is a fourth generation antibiotic, both of which are used to treat multi-drug resistant strains. 

20) P.17, line 12: Fig. 6: the resolution of the figure is too low and this does not allow to verify the data. What is the control of experiment? the untreated strains? the images of the strains treated only with cefixime are missing.

Author Response: The experiment has been revised as per instruction of the reviewers. We have performed the experiment along control as well to check the efficacy of our compound in treated and untreated way against the clinical isolates in the revised manuscript.

21) P.18, line 3: “Results obtained from the β- lactamase assay suggest that”…. Please indicate the total number of compounds similar to compound 11 found (28)

Author Response: The total numbers of compounds similar to compound 11 have been incorporated in the text (P.13 line 257). 

22) P.18, line 4: “having structural similarity”: is the chemical structure of the compounds shown in Table S1? if yes, please refer to the above table.

Author Response: Thank you for pointing out this. The corresponding table has been referred in the text. 

23) The Discussion section is a bit long and it would also benefit from some condensation.

Author Response: A condensation has been tried but some new experiments have also been performed to address comments of Reviewer 1 and Reviewer 2. So the addition of those results prevented the reflection of possible condensation. 

24) P.24, line 20: In my opinion, the sentence “The complete protocol used in this study is shown in Fig 8“ should be moved to the beginning of the results, in this way the experimental procedure is immediately clear to the reader of the manuscript.

Author Response: The figure 8 has been replaced as figure 1 in the text of the revised manuscript.

25) P.29, line 4: The method described for the β-lactamase activity assay is not clear. The authors should describe in detail the method used. Which sample is used in the assay? bacteria? which bacterial strains? what are the conditions used to grow bacteria?

Author Response: The authors have provided the brief explanation of the protocol regarding beta-lactamase activity assay in the revised manuscript on P.25 line 514-524. Sample in the assay have been mentioned (Chembridge compounds). No bacterial samples were used in this assay. 

26) P.29, line 15: describe punctually the protocol

Author Response: The schematic flow of complete protocol has been given in Figure 1. 

27) P.29, line 16: did the authors characterize (at the genomic level) the clinical strains of Enterobacter used? do these strains contain the gene that encodes for CMY-10? why did the authors not test Enterobacter ATCC reference strains? Please, perform these experiments.

Author Response: The experiments have been performed again as instructed by the reviewers in order to test Enterobacter ATCC reference strains page 26 line 540-541. Molecular identification assay has additionally been performed to characterize clinical strains of Enterobacter at genomic level while addressing the overall revisions at page 26 line 525-539.

28) P.29, line 22: Change the sentence: “The test was run for 24 hrs at 37 °C” with the sentence: “The plates were incubated for 24 hrs at 37 °C”

Author Response: The sentence has been revised accordingly. 

29) P.31, line 4: Table S1?????? The order of Supplementary Table is wrong.

Author Response: The order of all the supplementary tables has been revised.

---

## [Decision Letter · Decision Letter 1]

21 Dec 2020

Discovery of beta-lactamase CMY-10 inhibitors for combination therapy against multi-drug resistant Enterobacteriaceae

PONE-D-20-14041R1

Dear Dr. Syed Sikander Azam,

We’re pleased to inform you that your manuscript has been judged scientifically suitable for publication and will be formally accepted for publication once it meets all outstanding technical requirements.

Kind regards,

Massimiliano Galdiero, M.D., Ph.D.

Academic Editor

PLOS ONE

Additional Editor Comments (optional):

Reviewers' comments:

Reviewer's Responses to Questions

**Comments to the Author**

1. If the authors have adequately addressed your comments raised in a previous round of review and you feel that this manuscript is now acceptable for publication, you may indicate that here to bypass the “Comments to the Author” section, enter your conflict of interest statement in the “Confidential to Editor” section, and submit your "Accept" recommendation.

Reviewer #1: All comments have been addressed

Reviewer #2: All comments have been addressed

Reviewer #3: All comments have been addressed

2. Is the manuscript technically sound, and do the data support the conclusions?

Reviewer #1: Yes

Reviewer #2: Yes

Reviewer #3: Yes

3. Has the statistical analysis been performed appropriately and rigorously? 

Reviewer #1: Yes

Reviewer #2: Yes

Reviewer #3: N/A

4. Have the authors made all data underlying the findings in their manuscript fully available?

Reviewer #1: Yes

Reviewer #2: Yes

Reviewer #3: Yes

5. Is the manuscript presented in an intelligible fashion and written in standard English?

Reviewer #1: Yes

Reviewer #2: Yes

Reviewer #3: Yes

6. Review Comments to the Author

Reviewer #1: THIS PAPER IS CLEARLY REVISIONED FOLLOWING REVIEWER' SUGGESTIONS. I BELIEVE THAT AFTER THIS REVISION THE PAPER IS SUITABLE FOR PUBBLICATION.

Reviewer #2: Authors should pay attention to the nomenclature of bacteria throughout the text including the tables (see Table 2). The bacterial species must always be written in lowercase italics.

Furthermore, the units of measurement reported throughout the text must be checked (see Table 4 and related text).

Finally, in the Molecular assay, as in the electrophoresis there is no control it would be useful to insert in the text a reference on the high molecular weight plasmid.

Reviewer #3: (No Response)

7. PLOS authors have the option to publish the peer review history of their article (what does this mean?). If published, this will include your full peer review and any attached files.

Reviewer #1: No

Reviewer #2: No

Reviewer #3: No

---

## [Editor Report · Acceptance letter]

7 Jan 2021

PONE-D-20-14041R1 

Discovery of beta-lactamase CMY-10 inhibitors for combination therapy against multi-drug resistant Enterobacteriaceae 

Dear Dr. Azam:

I'm pleased to inform you that your manuscript has been deemed suitable for publication in PLOS ONE. Congratulations! Your manuscript is now with our production department. 

Kind regards, 

on behalf of

Prof. Massimiliano Galdiero 

Academic Editor

PLOS ONE